# Characterising open chromatin in chick embryos identifies cis-regulatory elements important for paraxial mesoderm formation and axis extension

Gi Fay Mok[1], Leighton Folkes [1], Shannon A. Weldon[1], Eirini Maniou [1,3], Victor Martinez-Heredia[1], Alice M. Godden [1], Ruth M. Williams[2], Tatjana Sauka-Spengler [2], Grant N. Wheeler [1], Simon Moxon [1] & Andrea E. Münsterberg [1✉]

Somites arising from paraxial mesoderm are a hallmark of the segmented vertebrate body plan. They form sequentially during axis extension and generate musculoskeletal cell lineages. How paraxial mesoderm becomes regionalised along the axis and how this correlates with dynamic changes of chromatin accessibility and the transcriptome remains unknown. Here, we report a spatiotemporal series of ATAC-seq and RNA-seq along the chick embryonic axis. Footprint analysis shows differential coverage of binding sites for several key transcription factors, including CDX2, LEF1 and members of *HOX* clusters. Associating accessible chromatin with nearby expressed genes identifies cis-regulatory elements (CRE) for *TCF15* and *MEOX1*. We determine their spatiotemporal activity and evolutionary conservation in Xenopus and human. Epigenome silencing of endogenous CREs disrupts *TCF15* and *MEOX1* gene expression and recapitulates phenotypic abnormalities of anterior–posterior axis extension. Our integrated approach allows dissection of paraxial mesoderm regulatory circuits in vivo and has implications for investigating gene regulatory networks.

[1] School of Biological Sciences, Cell and Developmental Biology, University of East Anglia, Norwich Research Park, Norwich, UK. [2] MRC Weatherall Institute of Molecular Medicine, Radcliffe Department of Medicine, University of Oxford, Oxford, UK. [3] Present address: Developmental Biology and Cancer, University College London, Great Ormond Street Hospital, Institute of Child Health, London, UK. ✉email: a.munsterberg@uea.ac.uk

The partitioning of paraxial mesoderm into repetitive segments, termed somites, is a key feature of vertebrate embryos. During amniote gastrulation, mesoderm cells emerge from the primitive streak and migrate in characteristic trajectories to generate axial, paraxial and lateral plate mesoderm (LPM)[1,2]. Paraxial mesoderm is located on either side of the midline tissues, neural tube and notochord. As the body axis extends, it consecutively generates pairs of somites[3] epithelial spheres comprised of multipotent progenitor cells. In response to extrinsic signals, epithelial somites (ES) undergo dramatic morphogenetic changes and reorganise[4–7]. On the ventral side cells undergo an epithelial to mesenchymal transition (EMT) to form the sclerotome, while on the dorsal side the cells in the dermomyotome remain epithelial. From the dermomyotome edges cells transition to form the myotome, in-between the sclerotome and dermomyotome[8]. Concomitantly with somite morphogenesis, the differentiation potential of somite cells becomes more restricted, with cells eventually becoming specified towards the lineages of the musculoskeletal system, including chondrocytes and skeletal muscle cells[4]. Overall the process of somitogenesis generates a spatiotemporal gradient of differentiation within the paraxial mesoderm along the embryonic body axis[3].

In addition, somite derivatives exhibit regional differences depending on their anterior–posterior axial position. Regional identity is already established at gastrula stages and is controlled by the stepwise transcriptional activation of HOX gene expression[9–11]. For example, members of the HOXB cluster are first activated in a temporal colinear fashion in prospective paraxial mesoderm, prior to ingression through the primitive streak[12]. The colinear activation of HOX genes culminates in nested expression domains within the paraxial mesoderm, thereby conferring regional identity along the axis[13,14]. To determine the structural features associated with colinear expression, the 3D organisation of HOX clusters has been investigated[10]. It has also been shown that posterior Wnt signalling and CDX transcription factors (TFs) are important regulators of the "trunk" HOX genes in the centre of HOX clusters[15]. In particular, CDX2 is essential for axial elongation with mutations leading to posterior truncations associated with changes in HOX expression domains[16]. CDX activity is associated with histone acetylation and mediates chromatin accessibility of regulatory elements[17].

Superimposed onto regional differences is the control of cell identity and differentiation, and several well-characterised TFs serve as markers for musculoskeletal lineages. Chondrogenic cells express PAX1, PAX9 and SOX9 and dermomyotomal myogenic progenitors are characterised by PAX3 and PAX7. Committed myoblasts express MYF5 and MYOD, while MYOG and KLHL31 are markers for differentiated myocytes[18–20]. Other transcriptional regulators that are important in paraxial mesoderm include TCF15 (Paraxis), a bHLH TF required for somite epithelialization;[21] CDX (Caudal), which is necessary for axis elongation;[22] and MEOX1, which is involved in somite morphogenesis, patterning and differentiation, particularly of sclerotome-derived structures[23,24]. In human, mutations of MEOX1 are found in patients with Klippel-Feil Syndrome, which is associated with fusion and numerical defects in the cervical spine as well as scoliosis[25,26]. Whilst the sequence of marker gene expression in paraxial mesoderm is well defined[19,20], the epigenetic and genomic mechanisms that control these transcriptional programmes remain largely unknown. The identification of enhancers has improved through high-throughput sequencing assays and comparative genomic analysis, however, experimental validation of enhancer activity remains challenging. In this study, we assay spatiotemporal changes in both gene expression signatures and accessible chromatin that occur in differentiating paraxial mesoderm along the anterior–posterior axis. We define differentially accessible chromatin regions within HOX genes that are associated with regional identities. Footprint analysis shows differential occupancy and coverage of binding sites along the axis for several TFs, including HOXA10, HOXA11, CDX2, LEF1 and RARA. CDX2 and LEF1 are both involved in similar processes during axis extention. However, network analysis shows that CDX2 and LEF1 footprints are associated with different expressed genes and there is little overlap in the genes they interact with. Correlating accessible chromatin with nearby expressed genes identifies cis-regulatory elements (CREs). We focus here on enhancers located upstream of TCF15 and MEOX1 and validate these in vivo, using electroporation of fluorescent reporters into gastrula-stage chick embryos. Time-lapse imaging shows the onset of enhancer activation in paraxial mesoderm and mutation of candidate TF motifs or epigenome modification leads to loss of gene expression and phenotypic changes. The MEOX1 CRE is evolutionary conserved in amphibians and human. Altogether our data characterises the accessible chromatin and gene expression landscapes in paraxial mesoderm, at different stages of somite maturation.

## Results

**Transcriptional profiling of developing paraxial mesoderm.** To conduct genome-wide transcriptome analysis during the spatiotemporal transition of paraxial mesoderm, we collected presomitic mesoderm (PSM), ES, maturing somites (MS) and differentiated somites (DS) from Hamburger–Hamilton stage 14 (HH14)[27] chick embryos in triplicate (Fig. 1a). At this stage, the four most posterior somites are epithelial, but in MS cells in the ventral part undergo EMT, the dorsal dermomyotome lip forms in the epaxial domain adjacent to the neural tube and myogenic cells begin to transition into the early myotome. Differentiating somites are compartmentalised, with a primary myotome beneath the dermomyotome and a sclerotome ventrally[5,28].

After harvesting, tissues were processed for RNA sequencing (RNA-seq) (Fig. 1). Principal component analysis (PCA) showed that PSM, ES, MS and DS samples cluster into three distinct groups, with MS and DS samples clustering together (Supplementary Fig. 1a). Differential gene expression analysis comparing PSM and ES revealed up-regulation of 713 genes and down-regulation of 583 genes; comparing ES and MS revealed up-regulation of 145 genes and down-regulation of 155 genes; and comparing MS and DS revealed up-regulation of 53 genes and down-regulation of 26 genes. Comparisons between samples confirmed that the greatest differential was observed between PSM and any of the somite samples, followed by the number of differentially expressed genes between the most recently formed epithelial somites and the most differentiated somites (ES versus DS) (Supplementary Fig. 1b).

Previously described somite TFs, such as NKX6-2, NKX3-2, ZIC1 and HES5, were highly enriched in ES compared to PSM, as well as the gap junction protein GJA5 (Connexin 40). Marker genes important for myogenic (MyoD1 and ACTC1) and chondrogenic (PAX9, FST) cell lineages were enriched in MS compared to ES. Markers for chondrocytes (Chondromodulin, CNMD), bone homoeostasis (Leucine-rich repeat containing, LRRC17) and cartilage (Keratan sulfate proteoglycan Keratocan, KERA) were identified (Fig. 1b–d). Myogenin (MYOG), a TF involved in differentiation of muscle fibres was enriched in DS compared to MS, as was expression of the neural crest cell (NCC) TF SOX10, due to NCCs migrating through the rostral half of differentiating somites. Other genes highly expressed in DS include the serine protease inhibitor, SPINK5; Troponin T2 (TNNT2) and Myomesin (MYOM1), encoding important proteins of the contractile sarcomere; CREBRF, a negative regulator

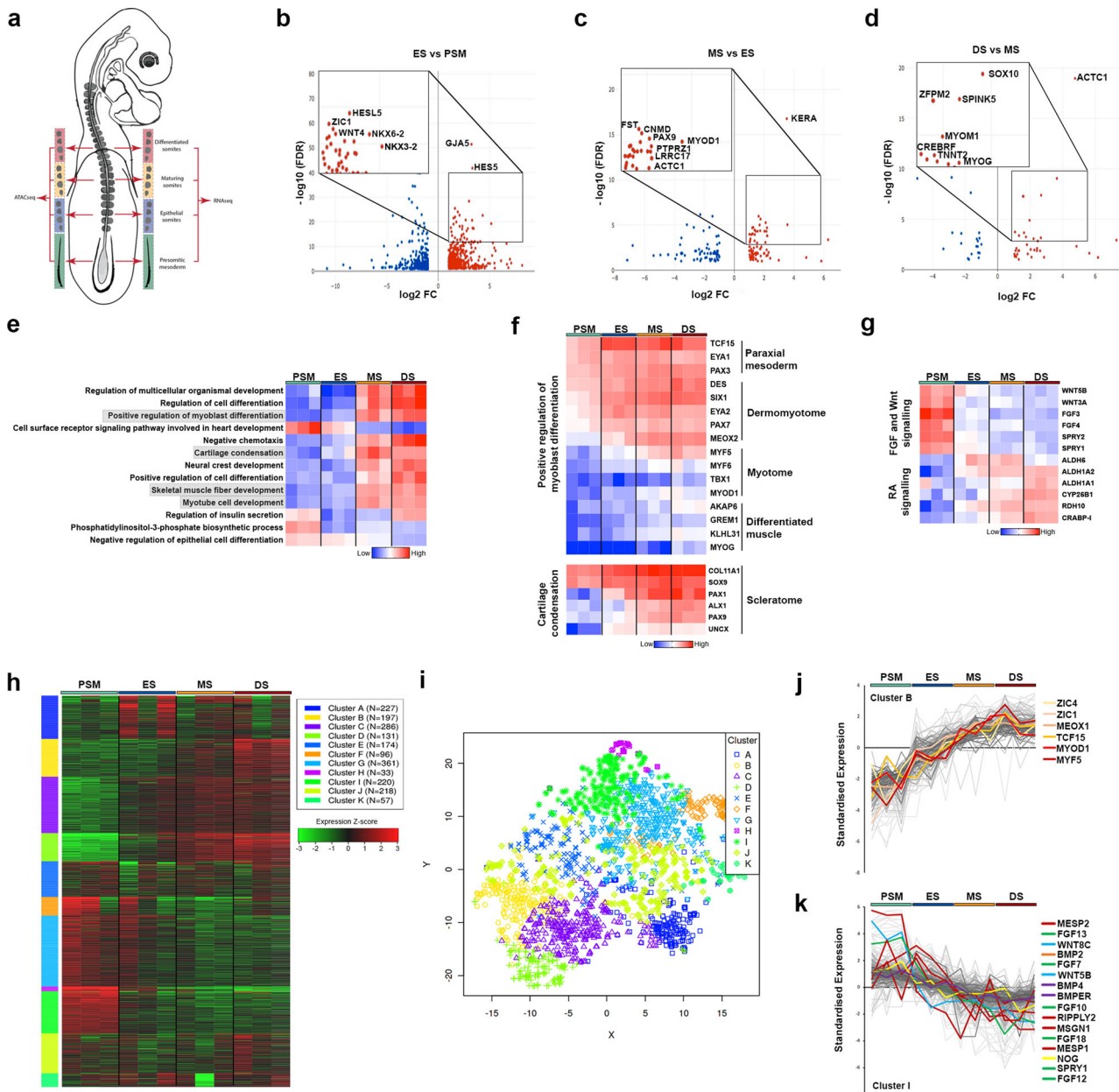

**Fig. 1 Transcriptional profiling of developing somites. a** Schematic representation of HH14 chick embryo with presomitic mesoderm (PSM), epithelial somite (ES), maturing somite (MS) and differentiated somite (DS) dissected for RNA-seq and ATAC-seq, in triplicate. **b–d** Volcano plots showing enriched genes (log fold change >1.5) comparing PSM with ES, ES with MS, and MS with DS. **e** Heat map showing GO terms associated with PSM or DS enriched genes. **f** Clusters of highly correlated genes identified for myoblast differentiation and cartilage condensation. **g** Wnt, FGF and retinoic acid (RA) signalling pathways are shown in heat map. **h** Heat map showing k-means linear enrichment clustering across PSM, ES, MS and DS. **i** Dimension reduction algorithm t-SNE used to map top genes in each cluster determined by k-means. **j** Cluster B and **k** Cluster I are shown with some genes labelled to identify key transcription factors and signalling components.

of the endoplasmic reticulum stress response and *ZFPM2*, a zinc finger TF (Fig. 1d).

The functional clustering by gene ontology (GO) terms of differentially expressed genes across all four stages reveals enrichment of biological processes involved in cell differentiation in DS and MS versus PSM and ES samples. Genes involved in myoblast differentiation, cartilage condensation, skeletal muscle fibre development and myotube cell development were up-regulated (Fig. 1e). Further analysis of genes involved in positive regulation of myoblast differentiation shows that they display dynamic expression across the four groups and include classic markers for different stages of paraxial mesoderm

differentiation. Genes differentially expressed in somite compartments include in the dermomyotome and myotome: *MYF5*, *MYF6* and *MYOD1*, whilst *MYOG* and *KLHL31* are associated with differentiated muscle. Classic markers for chondrogenesis and cartilage condensation within the sclerotome include the TFs, SOX9, PAX1 and PAX9, and the extracellular matrix component, COL11A1 (Fig. 1f). Functional clustering of differentially expressed genes also reveals enrichment of signalling pathways involved in anterior–posterior pattern formation. These pathways are expressed in an opposing fashion and include the FGF and Wnt signalling pathways, which are highly expressed in PSM and the retinoic acid (RA)

signalling pathway, which is more highly expressed in somite samples (Fig. 1g).

We next used weighted gene co-expression network analysis[29,30] to characterise gene co-expression clusters across the four samples of the top 400 differentially expressed genes. We identified 11 clusters based on k-means clustering. The heat map shows the gene expression levels across the four different samples, and the t-SNE plot illustrates the dimensional distribution of the different clusters (Fig. 1h, i). The top three GO terms associated with clusters include "anatomical structure morphogenesis" (Supplementary Fig. 1f). Clusters B and I comprise genes that increase or decrease in expression across the spatiotemporal series, from PSM to DS (Fig. 1j, k). Cluster I features components of FGF (*FGF13*, *FGF7*, *FGF10*, *FGF18*, *SPRY1*), BMP (*BMP2*, *BMP4*, *BMPER*, *NOG*) and WNT (*WNT8C*, *WNT5B*) signalling pathways in addition to classic PSM markers such as *MESP2*, *RIPPLY2*, *MSGN1* and *MESP1*, known to be important for somitogenesis. Cluster B features markers of cellular differentiation programmes such as *ZIC1*, *ZIC4*, *MEOX1*, *TCF15* and include the myogenic regulatory factors, *MYOD1* and *MYF5*.

**Profiling chromatin accessibility dynamics in paraxial mesoderm along the anterior–posterior axis.** To identify genomic regulatory elements that control paraxial mesoderm and somite differentiation programmes, we used Assay for Transposase-Accessible Chromatin using sequencing (ATAC-seq)[31]. This mapped chromatin accessibility across the paraxial mesoderm along the axis, in PSM, ES, MS and DS (Fig. 1a). Distinct chromatin accessibility profiles were evident at different stages of somite development, indicative of the dynamic progression of axial development. PCA showed a high reproducibility between biological triplicates of each sample type (Supplementary Fig. 2g–l), but dynamic changes in chromatin accessibility were observed between them. Using DiffBind[32,33] we show the densities and clustering of differentially accessible chromatin regions (peak sites) (rows), as well as the sample clustering (columns) for PSM against ES, ES against MS, and MS against DS. We identified differentially accessible peaks with differential densities showing clusters of peak sites with distinct patterns of chromatin accessibility levels for PSM against ES (Fig. 2a), ES against MS (Fig. 2b), and MS against DS (Fig. 2c). MA plots show the highest number of differentially accessible peaks is evident when comparing PSM and ES ($n = 27,692$, Fig. 2d). The number of differentially accessible peaks is lower when comparing ES against MS, and MS against DS ($n = 4670$, $n = 1965$, Fig. 2e, f). This is in line with the transcriptome data, where greater differences were seen between PSM and ES compared to the differences observed between different stages of somite maturation.

The genomic distribution of accessible regions was similar in all four sample types: between 39 and 42% were in intergenic regions, ~10% were in introns, 0.5% in exons, 2–3% at the TSS and 43–46% of accessible regions were within a 50 kb region upstream of the TSS which includes the promoter (Fig. 2g). Functional terms associated with predicted TF binding sites that were enriched in accessible peaks in DS compared to PSM included cell fate specification and terms related to morphogenesis or skeletal myogenesis (Fig. 2h). Consistent with the latter, we identified >200 binding sites for myogenin (MYOG) that are located within accessible chromatin peaks within 2 kb of genes differentially expressed in DS, where skeletal muscle differentiation occurs (Fig. 2i). The MYOG motif is well conserved across mouse and human, thus is likely to be conserved across avian species also. Other enriched TFs identified include bHLH proteins (TCF12, ASCL1, ARNT1), of which TCF12 is expressed in skeletal muscle, is part of the canonical Wnt pathway and

implicated as a transcriptional repressor in colorectal cancer[34]. Specificity proteins, Sp1, Sp2, Sp3 and Sp8, are zinc finger proteins known to interact with bHLH proteins such as MyoD[35]. Furthermore, Sp8 is a downstream effector of the Wnt pathway in neuromesodermal stem cells[36]. Sp1 and Sp3 bind to GC and GT boxes and can be displaced from these sequences by KLF16, a Krüppel-like zinc finger protein for which binding sites are also enriched and increased in PSM (Fig. 2j). Other zinc finger TFs include ZNF384, ZNF740 and ZNF263, which are involved in the regulation of cell differentiation genes including those relevant to musculoskeletal development. For example, ZNF384 regulates extracellular matrix genes MMP1, MMP3, MMP7 and COL1A1[37]; ZNF740 recruits the chromatin regulator HDAC1 to the SMAD4-DNA complex and prevents the recruitment of the transcriptional activators CREBBP and EP300[38]. The binding motifs for ZNF740 are increased in PSM (Fig. 2j). ZNF263 is involved in adipogenesis[39]. The Ewing sarcoma RNA binding protein 1 (EWSR1) regulates gene expression, cell signalling, RNA processing and transport. Chimeric proteins resulting from chromosomal translocations between EWSR1 and various TF genes[40,41] are involved in tumorigenesis such as Ewing sarcoma in bones and bone connective tissues. Furthermore, the binding motif for retinoic acid receptors (RXRA) is enriched. Motif enrichment analysis of differentially accessible regions identified additional TF motifs that were increased in number in either PSM (Fig. 2j) or DS (Fig. 2k). In PSM this included motifs for TFAP2C/TFAP2B and ZIC3/ZIC4, whose functions in axial elongation and/or musculoskeletal development are currently unknown. In DS, this included motifs for FOXO1/FOXO3 and MEOX1.

**Identification of differential footprints during somite development.** To further interrogate the accessible chromatin landscape during somite development, we used HINT-ATAC[42] to discover differential TF footprints in regions of open chromatin identified in PSM, ES, MS or DS. Initially we focussed on the CDX2 TF, which is a readout for posterior WNT signalling and has been implicated in defining neuromesodermal progenitors (NMP)[43]. CDX2 is essential for axial elongation[22] and is highly expressed in the PSM (Fig. 3a). Consistent with high levels of WNT signalling activity in the PSM, HINT-ATAC identified a greater number of CDX2 footprints in open chromatin in this region when compared to ES, MS and DS (Fig. 3b–d). Similarly, LEF1, a transcriptional effector for canonical WNT signalling, is highly expressed in the PSM (Fig. 3e). LEF1 is also expressed in somites, where it becomes restricted to the myotome[44,45]. We identified a greater number of LEF1 footprints in the PSM when compared to any of the somite samples, consistent with the more restricted expression of *LEF1* in the latter (Fig. 3f–h).

A reverse coverage pattern was observed for TFs involved in somite differentiation. For PAX3, an important TF that regulates the myogenic programme and highly expressed during somite development, HINT-ATAC revealed an increase in the number of PAX3 footprints in ES open chromatin when compared to PSM. The number of PAX3 footprints increased further in MS and DS when compared to PSM (Supplementary Fig. 3a–d) suggesting there is a greater coverage of bound sites in maturing and differentiating somites consistent with the role of PAX3 in myogenic progenitors in the dermomyotome. Another key somite TF, TWIST2 (also known as DERMO1), is important for EMT during somitogenesis. *TWIST2* was highly expressed in the paraxial mesoderm and expression increased as somites differentiate (Supplementary Fig. 2e). The number of genome-wide TWIST2 footprints were very similar in PSM and ES

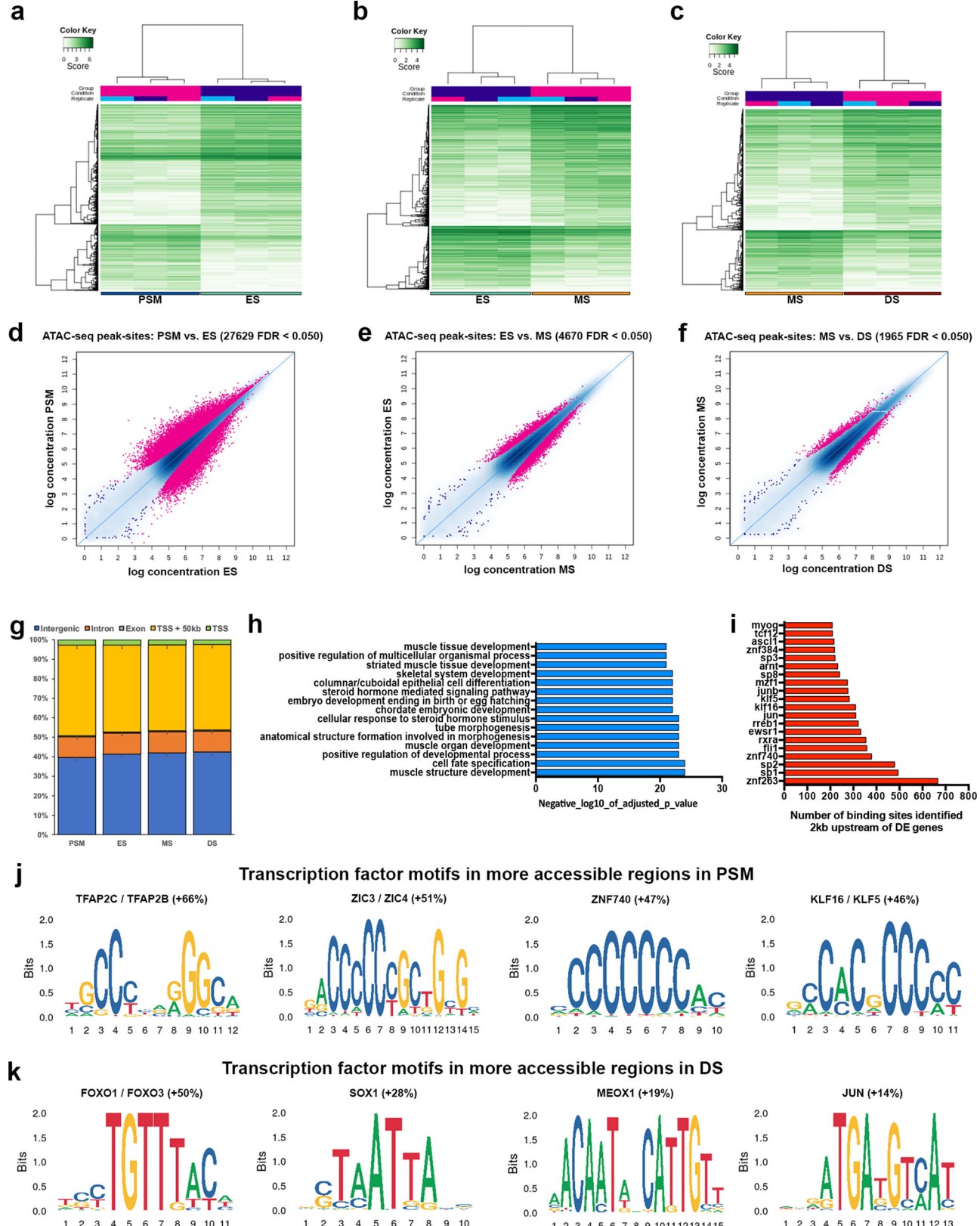

(Supplementary Fig. 2f), however, the number of footprints increased in MS and DS when compared to PSM (Supplementary Fig. 2g, h). Retinoic acid receptor alpha (RARA) is a nuclear receptor, which acts as a transcriptional repressor in absence of ligand but a transcriptional activator when RA is present (see[46] for review). RARA is highly expressed in somites (Supplementary Fig. 3i). HINT-ATAC identified fewer RARA footprints in PSM compared to MS and DS (Supplementary Fig. 3k, l).

The inverse coverage patterns identified for CDX2, LEF1 versus RARA were consistent with the opposing expression patterns for WNT and RA pathway components within paraxial mesoderm along the anterior–posterior axis (Fig. 1g). To further dissect the

**Fig. 2 Genome-wide profile of chromatin accessibility dynamics during somite development. a** Correlation heat maps of accessible chromatin regions (ATAC-seq peak sites) comparing PSM and ES, **b** ES and MS and **c** MS and DS. **d** MA plots of significantly differential peak sites (pink) comparing PSM with ES, **e** ES with MS and **f** MS with DS. **g** Bar plot showing proportions of total genome sequence of peaks in PSM, ES, MS and DS (error bars = SD). Nearly half of all peaks lie within 50 kb of the promoter and TSS and half are in intergenic and intron regions. **h** GO terms associated with enriched transcription factors in DS compared to PSM. For enriched GO terms, p values were obtained from a modified Fisher exact test. **i** Number of transcription factor binding sites identified within 2 kb upstream of differentially expressed genes in DS compared to PSM. **j** Motif enrichment analysis performed using Homer. Transcription factor motifs identified as enriched in PSM accessible regions and **k** in DS accessible regions. Percentage indicates increase of motifs identified in either PSM or DS for each motif.

roles of CDX2 and LEF1 in posterior axis elongation we determined genes associated with either CDX2 or LEF1 footprints in accessible regions within 10 kb upstream or downstream. GO terms for these genes were overlapping and include: anatomical structure morphogenesis/development, metabolic process and regulation, for both CDX2 (Fig. 3i) and LEF1 (Fig. 3j). We next performed STRING analysis, using a threshold of 0.700, to obtain a protein–protein interaction (PPI) map for genes identified with CDX2 (Fig. 3k) or LEF1 (Fig. 3l) footprints in accessible regions within 10 kb. This revealed genes with strong PPIs, including those associated with enriched biological processes such as embryonic morphogenesis for CDX2 and animal organ morphogenesis for LEF1. LEF1 footprints correlated with CDX2 consistent with CDX2 being regulated by the Wnt signalling pathway. Furthermore, the phenotypical traits of CDX2 mouse mutants[47] include posterior truncations reminiscent of those found in LEF1/TCF1 double mutants[48]. Thus, to explore whether CDX2 and LEF1 could regulate similar genes, we examined all differentially up-regulated genes in the PSM and investigated whether there are associated CDX2 and LEF1 footprints. We found 101 genes with a CDX2 footprint and 42 genes with a LEF1 footprint within 10 kb up- or downstream. Surprisingly, when comparing these sets of genes only four genes—*Msgn1, Sall4, Spry1 and DDC*—were associated with both CDX2 and LEF1 footprints, and the majority of correlated genes was different (Fig. 3m). Our analysis suggests that CDX2 and LEF1 are part of discrete networks acting in parallel to govern similar processes (Fig. 3k, l), but they regulate different sets of genes important for these processes.

**Chromatin accessibility and differential TF footprints in the HoxA cluster**. We next examined the *HOXA* cluster, one of four *HOX* gene clusters imposing regional identity along the anterior–posterior axis via the colinear expression of its members. We determined how *HOXA* gene expression patterns correlate with the accessible chromatin landscape. RNA sequencing determined expression levels of each member of the *HOXA* cluster in PSM, ES, MS and DS (Fig. 4a). Their expression reflects the organisation of the genes within the cluster: the more 3′ located genes have a more anterior expression boundary compared to the genes located more 5′, which are restricted more posteriorly. Accordingly, we find that *HOXA1, HOXA2, HOXA3, HOXA4, HOXA5* and *HOXA6* are all highly expressed across the length of the axis: in PSM, ES, MS and DS. A small decrease in *HOXA7* gene expression was detected in DS, with more pronounced decreases observed for *HOXA9, HOXA10, HOXA11* and *HOXA13*, which were also reduced progressively in MS and ES. The colinear pattern of gene expression correlated with differentially accessible chromatin within the *HOXA* cluster (Fig. 4b). Accessible chromatin regions were seen in PSM, ES, MS and DS near the promoter of *HOXA1, HOXA2, HOXA3, HOXA4, HOXA5* and *HOXA6*. However accessible chromatin for HOXA7 was reduced at the promoter in DS compared to PSM, ES and MS. For the more posteriorly restricted genes, *HOXA9, HOXA10, HOXA11* and *HOXA13* accessible chromatin peaks were reduced

in ES, MS and DS compared to PSM, which correlated with their reduced expression. We demonstrate the same relationship between gene expression and chromatin accessibility along the anterior–posterior axis across the *HOXB, HOXC* and *HOXD* clusters (Supplementary Fig. 4a–f). In the *HOXA* cluster we identified footprints within accessible regions in intergenic regions. Notably, we identified footprints for TFs involved in patterning along the anterior–posterior axis, including footprints for CDX1/2, LEF1 and for members of the HOX clusters themselves, as well as for some of the TFs with enriched motifs in accessible regions such as RXRA, TFAP2B/C, SP1, SP2, ZIC1/3, FOXO1/4, ZNF263 (Figs. 2i–k and 4b). To investigate the impact of the dynamic changes in *HOXA* gene expression along the anterior–posterior axis, we next explored the number of TF footprints for HOXA2, HOXA5, HOXA10 and HOX11 in PSM and DS (Fig. 4c–f). We observed the same number of footprints for HOXA2 and HOXA5 when comparing PSM and DS, however, a significant decrease in coverage was detected for HOXA10 and HOXA11 footprints in anterior DS compared to PSM. This reveals a strong association between gene expression levels along the anterior–posterior axis and the genome-wide coverage of HOXA binding sites.

**Identification and validation of paraxial mesoderm-specific regulatory elements**. Next, we identified differentially accessible peaks that were open specifically in PSM or in somite samples, ES, MS or DS. We hypothesise that these could represent putative enhancers. For example, differentially accessible peaks identified flanking genes highly expressed in the PSM included a peak downstream of *MSGN1* present in PSM and not in ES, MS or DS (Supplementary Fig. 5a); a peak downstream of *WNT8C* and a peak within intron 1 present in PSM but not in somite tissues (Supplementary Fig. 5b); and peaks upstream and downstream of *FGF4* present in PSM and low or absent in somite tissues (Supplementary Fig. 5c). For the muscle differentiation gene, *MYOG*, a peak was identified upstream of the gene in DS, MS and interestingly also in ES, but not in PSM (Supplementary Fig. 5d). For *RDH10*, which is associated with RA signalling and highly expressed in somites but less abundant in PSM, a differential peak was identified in ES, MS and DS and not in PSM (Supplementary Fig. 5e). Similarly for *GREM1*, an antagonist of BMP signalling highly expressed in developing somites, a differential peak present in all somite samples but not in PSM was identified downstream of the gene (Supplementary Fig. 5f). In most cases, chromatin accessibility correlated well with gene expression and in some cases it preceded transcript detection, e.g. *MYOG*, or high level gene expression, e.g. *TCF15* (see below). The putative enhancer activities of these differential peaks remain to be confirmed experimentally, however, we validated and further characterised some somite enhancers by embryo electroporation[49]. We focussed on *TCF15* and the homeodomain TF, *MEOX1*, two classic markers identified in the group of genes that increased during the differentiation of paraxial mesoderm and somites (Fig. 1j, Cluster B). In addition, MEOX1 binding motifs were increased in accessible regions in DS (Fig. 2k).

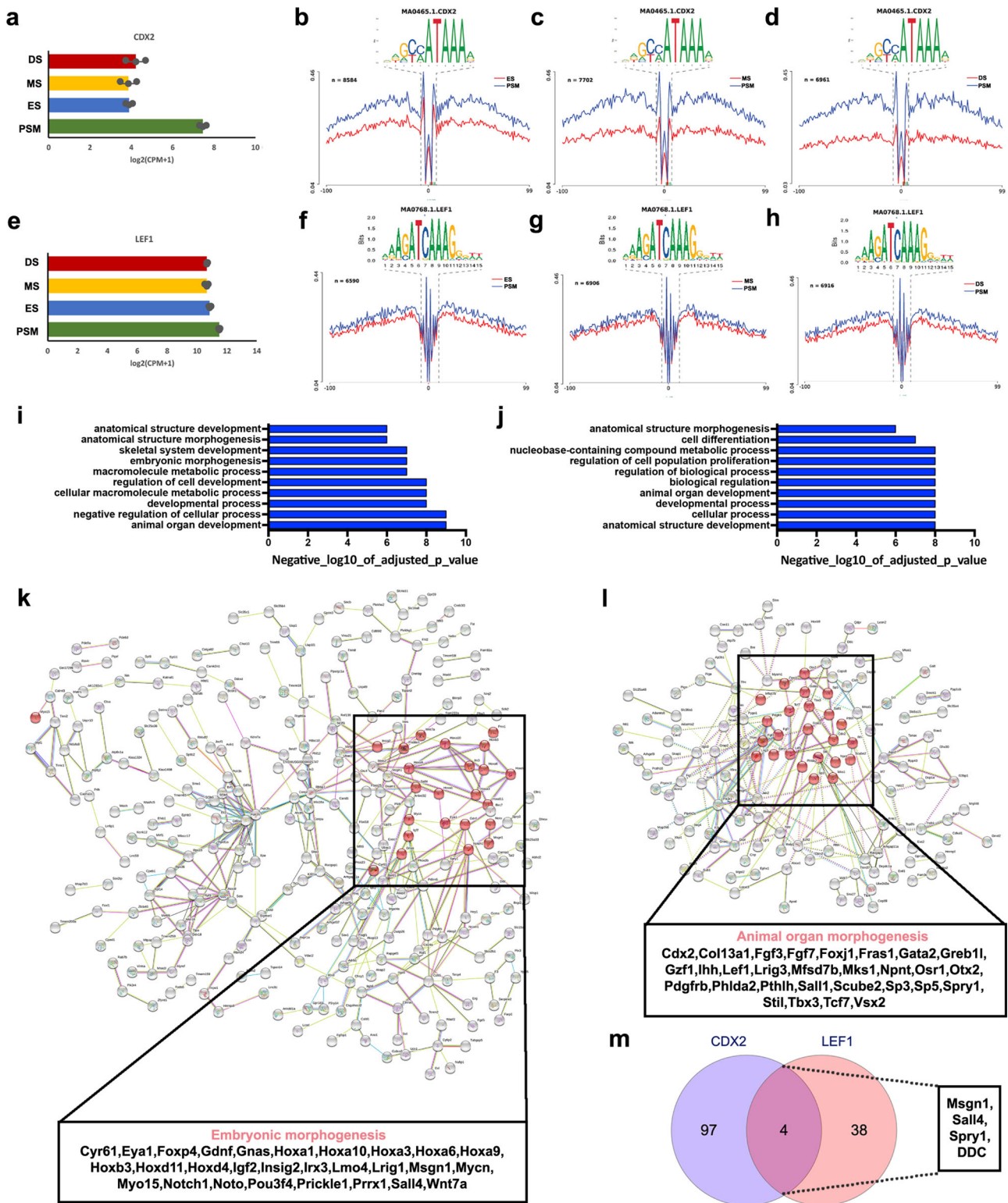

**Fig. 3 Differential footprints identified for CDX2 and LEF1 during somite development. a** Gene expression from mRNA-seq (error bars = SEM, n = 3) for *CDX2*. **b** Tn5 insertion frequency across all accessible regions containing at least one CDX2 motif, at nucleotide resolution in PSM, ES, MS and DS reveals the presence of a footprint centred on the CDX2 motif. Differential footprinting for CDX2 motif comparing PSM and ES, **c** PSM and MS, and **d** PSM and DS. **e** Gene expression for *LEF1* (error bars = SEM, n = 3). **f** Differential footprinting for LEF1 motif comparing PSM and ES, **g** PSM and MS, and **h** PSM and DS. Similar GO terms for genes associated with (**i**) CDX2 footprints and (**j**) LEF1 footprints. For enriched GO terms, *p* values were obtained from a modified Fisher exact test. **k**, **l** Protein–protein network analysis using STRING database. Interactions between genes identified with (**k**) CDX2 and (**l**) LEF1 footprints in an accessible region, within 10 kb upstream or downstream. Highlighted in red are genes correlated with embryonic morphogenesis in CDX2-associated genes and animal organ morphogenesis in LEF1-associated genes. **m** Venn diagram of CDX2-associated genes against LEF1-associated genes identified only four common genes—*Msgn1*, *Sall4*, *Spry1* and *DDC*.

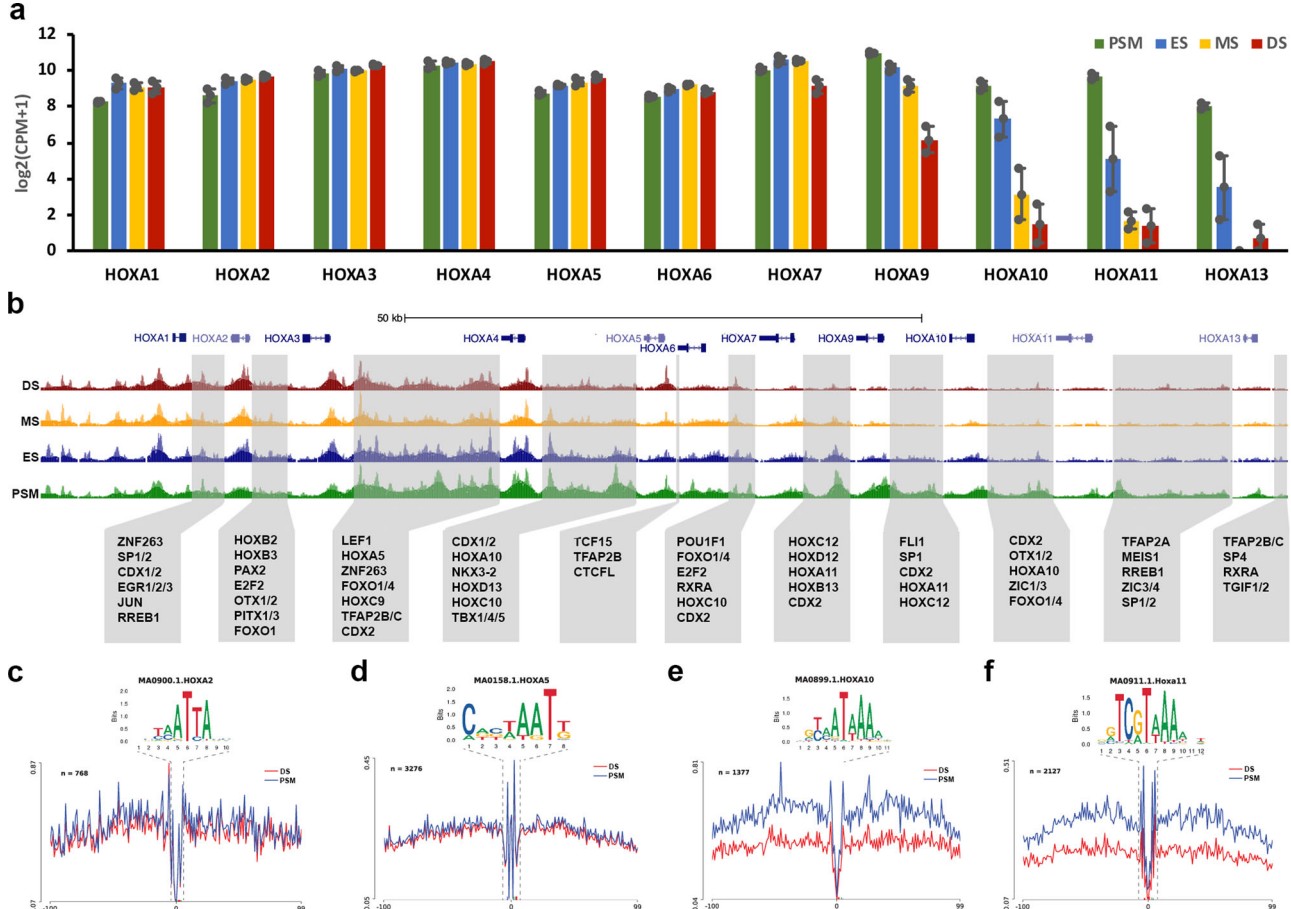

**Fig. 4 Chromatin accessibility and differential footprints for HoxA cluster. a** Gene expression from mRNA-seq for HoxA cluster (error bars = SEM, $n = 3$) in paraxial mesoderm regions. **b** Genome browser views of ATAC-seq profile across the *HoxA* cluster. ATAC and RNA profiles are shown in green for PSM, in blue for ES, in yellow for MS and in red for DS. Grey boxes indicate intergenic accessible regions and transcription factor footprints identified within those regions. Genome-wide differential footprinting for (**c**) HoxA2, (**d**) HoxA5, (**e**) HoxA10 and (**f**) HoxA11 between PSM and DS.

We examined open chromatin peaks flanking the *TCF15* and *MEOX1* genes within 10 kb. Identified peaks representing candidate CREs were cloned upstream of the herpes simplex virus thymidine kinase (HSV-TK) minimal promoter, driving expression of a stable Citrine reporter[50]. Electroporation targeted the prospective mesoderm of gastrula-stage HH3 + embryos (Supplementary Fig. 5g, h), and reporter gene expression profiles were monitored until HH11.

We identified two CREs upstream of *TCF15* (Fig. 5a). These sequences are chick specific and the fluorescent enhancer reporters showed spatially restricted activities. For the first element, *TCF15* Enh-1 (1500 bp) we observed activity in the PSM, in all somites and in the notochord (Fig. 5b). The second element, *TCF15* Enh-2 (700 bp), showed activity mainly in PSM and somites, as well as some activity in LPM (Fig. 5c, e). In situ hybridisation showed that expression of *TCF15* was restricted to PSM and somites (Fig. 5g)[51] and it is not clear at present why *TCF15* Enh-1 and *TCF15* Enh-2 drive ectopic reporter expression also in the notochord and LPM. To address the possibility that repressive elements that limit enhancer activity were missing, we combined *TCF15* Enh-1 and *TCF15* Enh-2. This reporter led to Citrine expression in PSM and LPM, however not in the notochord (Fig. 5d) suggesting that the region comprising *TCF15* Enh-2 may include elements that suppress ectopic expression in the notochord. It is possible that the TCF15 Enh-2 drives another gene in LPM cells, alternatively accumulation of Citrine in LPM may reveal sites of *TCF15* expression that cannot

be detected by in situ hybridisation. Time-lapse movies for *TCF15* Enh-2 show Citrine fluorescence was first detected in a HH6 embryo in prospective paraxial mesoderm cells as they converge towards the midline (Supplementary Fig. 5i and Supplementary Movie 1). Strong signal was seen in the first somite at HH7 and subsequently in all newly formed somites, as well as the PSM and prospective paraxial mesoderm cells.

Because reporter activity observed with *TCF15* Enh-2 reflected more closely the spatial gene expression pattern of *TCF15*, we next sought to identify TFs that regulate this element. HINT-ATAC identified a TF footprint for RARA within *TCF15* Enh-2, consistent with RARA expression and coverage of binding sites across the anterior–posterior axis (Supplementary Fig. 3i–l). Introducing mutations into the RARA binding site (Fig. 5a) led to loss of reporter activity in the embryo (Fig. 5f), suggesting RARA is indeed required to activate *TCF15* Enh-2. To determine the potential significance of RARA-mediated regulation of *TCF15* Enh-2 in vivo, we used the conventional dCas9-KRAB repressor to modify the endogenous enhancer[52]. Two CRISPR guide RNAs (gRNA) designed to target the repressor to the *TCF15* Enh-2 RARA binding site, or scrambled gRNA controls were electroporated together with dCas9-KRAB (Fig. 5g, h). Detection of *TCF15* expression by in situ showed that epigenome modification of the endogenous *TCF15* Enh-2 alone led to reduced *TCF15* expression and concomitantly a drastic truncation of the body axis ($n = 6/8$ embryos), whilst control scrambled gRNAs/dCas9-KRAB repressor has no effect on *TCF15* expression or axis

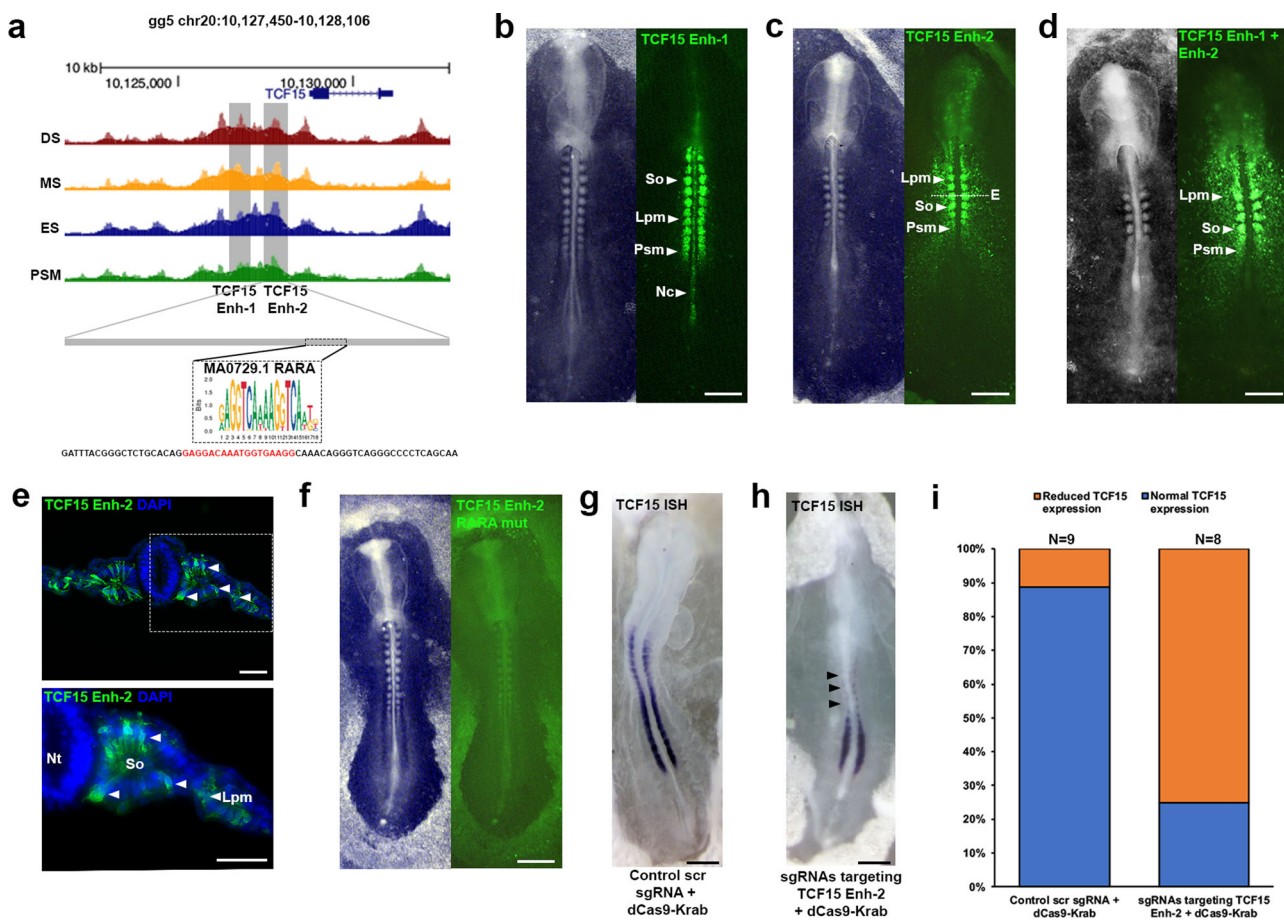

**Fig. 5 Identification of a regulatory element for TCF15. a** ATAC-seq profile at the *TCF15* locus. Grey boxes indicate putative enhancers identified (*TCF15* Enh-1 and *TCF15* Enh-2). RARA footprint identified within *TCF15* Enh-2. Mutant reporter sequence for *TCF15* Enh-2 RARA mutant. **b** *TCF15* Enh-1 (*n* = 15/15) and **c** *TCF15* Enh-2 (*n* = 9/9) reporter expression in presomitic mesoderm (Psm), notochord (Nc), somites (So) and lateral plate mesoderm (Lpm). **d** Combined *TCF15* Enh-1/Enh-2 reporter expression (*n* = 9/9). **e** Transverse sections of embryo in **c** immunostained for Citrine showing *TCF15* Enh-2 expression in somites and lateral plate mesoderm, white dashed line in **c** indicates location of section, representative of (*n* = 4/4). Nuclei stained with DAPI (blue). **f** *TCF15* Enh-2 Citrine reporter with RARA binding site mutation displays lack of expression in somites (*n* = 6/6). Epigenome engineering using dCas9-Krab with (**g**) control scrambled sgRNAs resulted in no change (*n* = 8/9) and (**h**) sgRNAs targeting endogenous RARA binding site led to loss of *TCF15* expression (*n* = 6/8) as shown by wholemount in situ hybridisation. **i** Percentage of embryos with normal (blue) or reduced (orange) *TCF15* in situ expression after electroporation of control scrambled sgRNA with dCas9-Krab or sgRNAs targeting *TCF15* Enh-2 RARA binding site with dCas9-Krab. All scale bars = 500 μm except for **e** scale bar = 100 μm.

elongation (*n* = 8/9 embryos, Fig. 5g–i). These data suggest that RA signalling is crucial for *TCF15* gene expression as RARA binding site perturbation led to loss of reporter activity and epigenome editing of the endogenous CRE resulted in disruption of anterior–posterior axis elongation. This is consistent with mouse mutants of *TCF15* or mutants affecting RA signalling, in which ES formation is disrupted and the embryonic axis truncated[21,53,54]. Whilst it has been shown that Wnt signalling is important for *TCF15* expression in early somites[55], there is no evidence of direct regulatory interactions. However, we cannot exclude the possibility that Wnt signalling, via LEF/β-catenin, contributes to *TCF15* expression potentially via a different CRE, which alone is not sufficient. It is also worth noting that *TCF15* Enh-1 and Enh-2 are not conserved across mammalian species.

MEOX1 is an important TF for early somite patterning and differentiation[23,24]. We examined accessible regions of chromatin and selected an element that is evolutionary conserved at sequence level between chicken, Zebra finch, American alligator, Chinese softshell turtle, lizard, human and mouse. We identified one candidate CRE of 1095 kb, ~1 kb upstream of *MEOX1* (Fig. 6a). This element displayed enhancer activity, with

expression of the Citrine reporter restricted to the PSM and all somites (Fig. 6b, c). Expression in PSM was unexpected as chromatin was not accessible at that stage. It is possible that the citrine reporter is missing some repressive elements that are present endogenously. Time-lapse movies reveal Citrine fluorescence, which was first detected in the prospective paraxial mesoderm cells of a HH6 embryo. At HH7 signal was detected in the first somite and subsequently in all newly formed somites, as well as the PSM and prospective paraxial mesoderm cells. Overall the pattern was consistent with *MEOX1* gene expression detected in situ (Fig. 6d, f and Supplementary Movie 2). We identified two TF footprints within the enhancer, one for FOXO1 and one for ZIC3 (Fig. 6a). We next determined their requirement for the activation of fluorescent reporter expression. Mutation of FOXO1 or ZIC3 sites individually had no effect and reporter activity was still observed (Supplementary Fig. 6a, b). However, mutation of both sites led to loss of reporter activity (Fig. 6e). This suggests both TFs are able to activate this CRE and either FOXO1 or ZIC3 alone is sufficient. To investigate the significance of this element, we modified the endogenous enhancer using four gRNAs to target the dCas9-KRAB repressor to the *MEOX1* Enh. Scrambled

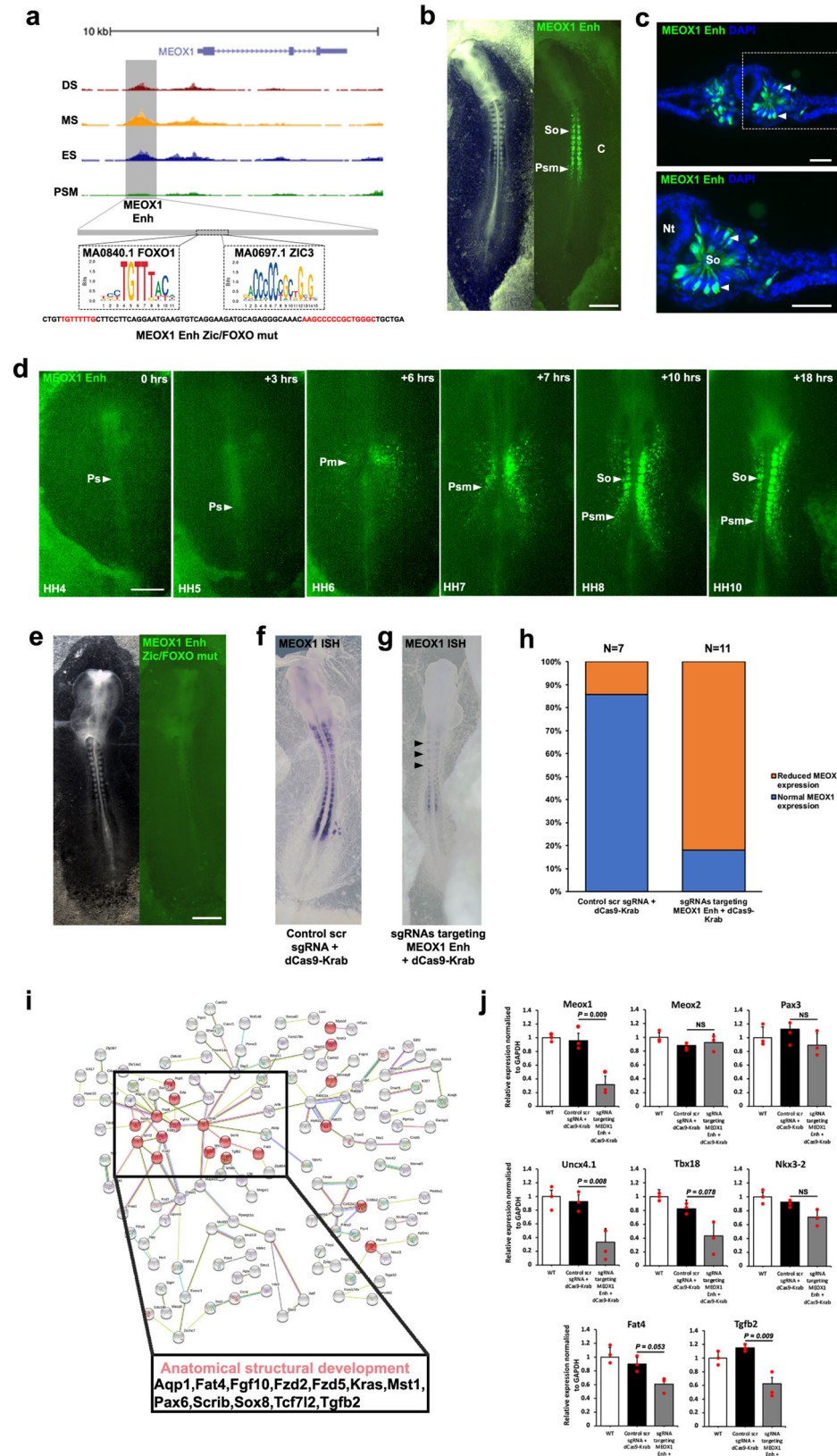

gRNAs with dCas9-KRAB were used as control (Fig. 6f–h). Using a probe to detect *MEOX1* transcripts showed that *MEOX1* Enh enhancer perturbation led to loss of gene expression (Fig. 6g) (*n* = 9/11). This was confirmed by RT-qPCR (Fig. 6j) and suggests the element is required.

To determine the genes potentially regulated by MEOX1 in paraxial mesoderm, we identified accessible chromatin regions within 10 kb of an expressed gene, which comprised a MEOX1 footprint (Supplementary Fig. 6c–e). STRING analysis of these putative MEOX1 regulated genes revealed PPI networks including

**Fig. 6 Identification of a regulatory element for MEOX1. a** ATAC-seq profile at *MEOX1* locus. Grey box indicates putative enhancer identified. FOXO1 and ZIC3 footprints identified within enhancer element. Mutant reporter sequence for *MEOX1* Enh ZIC/FOXO mutant. **b** *MEOX1* Enh reporter expression in presomitic mesoderm (Psm) and in somites (So) (*n* = 13/13). **c** Transverse sections of embryo in **b** immunostained for Citrine showing *MEOX1* Enh expression in epithelial somites, white dashed line in **b** indicates location of section, representative of (*n* = 4/4). Nuclei stained with DAPI (blue). **d** Still photographs from a time-lapse movie of *MEOX1* Enh with the primitive streak (Ps) indicated. Fluorescent activity first observed in prospective paraxial mesoderm (Pm) at HH6 and continuous expression in the presomitic mesoderm (Psm) at HH7 prior to expression in somites (So) at HH8 and HH10 (*n* = 3/3). **e** FOXO1 and ZIC3 binding site mutation in *MEOX1* Enh Citrine reporter led to loss of expression (*n* = 6/7). Epigenome engineering using dCas9-Krab with (**f**) control scrambled sgRNAs resulted in no change (*n* = 6/7) and (**g**) sgRNAs targeting endogenous FOXO1 and ZIC3 binding sites led to loss of *MEOX1* expression (*n* = 9/11) as shown by wholemount in situ hybridisation. **h** Percentage of embryos with normal (blue) or reduced (orange) MEOX1 expression after injection and electroporation of control scrambled sgRNA with dCas9-Krab or sgRNAs targeting MEOX1 Enh FOXO1 and ZIC3 binding sites with dCas9-Krab. **i** Protein–protein network analysis using STRING database. Interactions between genes identified with MEOX1 footprints in an accessible region, within 10 kb upstream or downstream. Highlighted in red are genes correlated with anatomical structural development based on GO analysis. **j** RT-qPCR on somites dissected from wild-type (WT) embryos, or embryos electroporated with control scrambled sgRNA and dCas9-Krab, or sgRNAs targeting *MEOX1* Enh FOXO1 and ZIC3 binding sites with dCas9-Krab. Statistical significance was determined by a two-tailed Student's *t* test. \*\**p* = 0.001–0.01; \**p* = 0.01–0.1; ns not significant. All scale bars = 500 μm except for **c** scale bar = 100 μm.

genes enriched with the GO term anatomical structural development, and also included components of signalling pathways, such as Wnt and TGFbeta (Fig. 6i). To confirm that some of these genes are involved in mediating the function of MEOX1 in paraxial mesoderm, we used RT-qPCR to assess their expression in normal and epigenome edited somites (Fig. 6j). First, we showed that *MEOX1* expression in wild-type somites was unaffected after introducing control gRNAs together with the dCAS9-KRAB repressor. However, electroporation of sgRNAs targeting the dCAS9-KRAB repressor to the *MEOX1* enhancer led to suppression of *MEOX1*. Expression of the closely related *MEOX2* gene was not affected, similarly *PAX3* expression remained unchanged. Next, we assessed a number of genes involved in chondrogenesis and setting up the polarity of the sclerotome. Epigenome editing of the endogenous *MEOX1* enhancer led to down-regulation of *Uncx4.1*, *TBX18*, *FAT4* and *TGFb2*, which are all associated with a MEOX1 footprint, indicating that they could be direct targets. Expression of *NKX3-2*, which is not associated with a MEOX1 footprint within 10 kb up- or downstream of the gene, was also inhibited after negative regulation of *MEOX1* by epigenome editing of the *MEOX1* enhancer.

As the *MEOX1* Enh is highly conserved amongst amniote taxa—birds, reptiles and mammals (Fig. 7a), we next asked whether the homologous mammalian sequences are active in chick. We found that a human *MEOX1* Enh, isolated from HeLa cells, was able to drive Citrine expression in somites. Activity was also detected in LPM and PSM (Fig. 7b). The human *MEOX1* Enh sequence included conserved FOXO1 and ZIC3 binding sites and mutation of both sites led to loss of reporter activity (Fig. 7c). Therefore, we propose transcriptional regulation of the *MEOX1* enhancer is highly conserved in human and chick, with FOXO1 and ZIC3 binding sites required for enhancer activity. Interestingly, the *MEOX1* Enh sequence was not found in fish or amphibians (Fig. 7a). However, when we injected the chick *MEOX1* Enh reporter into one cell of *Xenopus laevis* embryos at the 2-cell stage, we observed Citrine fluorescence in the paraxial mesoderm of early neurula stages (NF stage 14), where it overlapped with *MYOD* (Fig. 7d and Supplementary Fig. 7b). At NF stage 25, Citrine expression was detected in mesoderm and by NF stage 33 and stage 42 Citrine was visible in elongated muscle fibres, which are somite derived (Fig. 7d). The *MEOX1* Enh with mutations in the FOXO1 and ZIC3 binding sites showed no activity (Supplementary Fig. 7a). This suggests that in amphibians the *MEOX1* Enh can be activated by the same regulatory mechanism, even though the CRE is not conserved in the same location in the *Xenopus laevis* genome.

## Discussion

Extension of the vertebrate body axis is driven by the addition of new segments at the posterior end of the embryo. During avian gastrulation, presumptive paraxial mesoderm cells ingress through the primitive streak and their early migration patterns can be observed directly[1]. It has been shown that *HOX* genes are activated in a temporal colinear fashion, just prior to ingression when the precursors are still located in the epiblast[1,12], thus regional identity along the anterior–posterior axis is acquired at gastrula stages. Paraxial mesoderm formation continues as the main body axis forms[2] and cells are added from a bi-potential population of NMP cells found in the tailbud. In response to high levels of Wnt3a and CDX family members, these progenitors commit to mesoderm fates and give rise to neck, trunk and tail structures[14].

Here we provide detailed molecular profiles of paraxial mesoderm of cranial and trunk regions by RNA-seq and ATAC-seq[31]. Using samples from along the axis we identify differential gene expression signatures consistent with axial patterning and differentiation, including the appearance of chondrogenic and myogenic markers in more anterior differentiating somites (Fig. 1c, d, j)[5,8,19,28,51]. We show that genes correlated with cell fate specification and muscle development are enriched in differentiating somites, and that the TF binding motifs found upstream of differentially expressed genes include the motif for myogenin consistent with its role in myogenic differentiation (Fig. 2h, i). Components of signalling pathways involved in anterior–posterior axis patterning are also differentially expressed, such as FGF, Wnt and RA pathways (Fig. 1g, k)[53,56] and the motif for RXRA is enriched in differentiating somites (Fig. 2i). Furthermore, we uncover genome-wide dynamic changes in chromatin accessibility across the spatiotemporal series. Using HINT-ATAC, an improved method to predict TF binding sites with footprints[42], we show differential coverage of several binding sites along the anterior–posterior axis, including sites for RARA and LEF1, transcriptional effectors of the RA and Wnt pathways. Furthermore, we identify differential footprints for CDX2, a readout for WNT and a TF required for axial elongation (Fig. 3a–h and Supplementary Fig. 3j–l). The observed coverage patterns correlate well with gene expression and with the known functions of RA and Wnt signalling in anterior–posterior axis patterning. Interestingly, despite their similar function in posterior axis extension our network analysis shows that CDX2 and LEF1 footprints are largely correlated with different genes. Only four genes are associated with both CDX2 and LEF1 footprints, suggesting that posteriorization driven by Wnt signalling involves at least two parallel acting protein–protein networks. Three of the

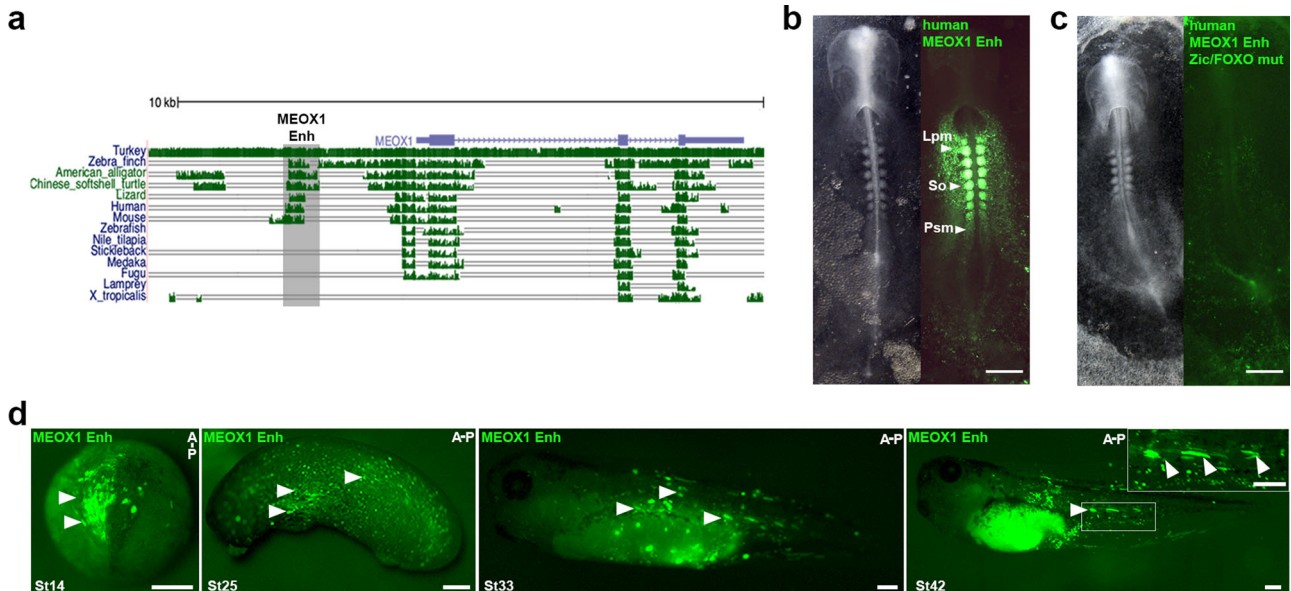

**Fig. 7 Evolutionary conservation of MEOX1 enhancer and regulatory mechanism across vertebrates. a** Genomic alignment of chick *MEOX1* locus, ~10 kb. Exons and introns are represented by blue boxes and lines. Clustered green vertical lines indicate sequence identity between different species. The height of bars indicates extent of conservation in different species: turkey, zebra finch, American alligator, Chinese softshell turtle, Lizard, Human, Mouse, Zebrafish, Nile tilapia, Stickleback, Medaka, Fugu, Lamprey and *Xenopus tropicalis*, as indicated on the left of each row. Grey shading indicates the location of the *MEOX1* Enh. **b** Expression of the conserved human *MEOX1* Enh reporter in presomitic mesoderm, somites and lateral plate mesoderm in a HH9 chick embryo (*n* = 6/6). **c** Mutation of FOXO1 and ZIC3 binding sites in the human *MEOX1* Enh Citrine reporter led to loss of expression (*n* = 4/4). **d** Chick *MEOX1* Enh reporter injected into 1 cell of a *Xenopus laevis* 2-cell embryo shows Citrine expression in paraxial mesoderm (St14, white arrowheads (*n* = 6/6)) and in early somites and elongated myofibres (St25 (*n* = 6/6), St33 (*n* = 6/6), St42 (*n* = 6/6), white arrowheads). A-P anterior–posterior. Scale bars for **b**, **c** = 500 µm, **d** scale bar = 250 µm.

shared proteins identified—Msgn1, Sall4 and Spry1 are involved in posteriorization. Msgn1 is a master regulator of paraxial mesoderm formation[57], Sall4 has recently been shown to regulate the balance between NMP maintenance and differention[58], and Spry1 is a negative feedback regulator of FGF signalling. Our finding that Msgn1, Sall4 and Spry1 are associated with both CDX2 and LEF1 footprints in PSM is highly consistent with their known function. The fourth shared gene, DCC (aromatic amino acid decarboxylase) is mutated in a rare genetic disorder and deficiency of this enzyme affects neurotransmitter production. However, its possible role in axial elongation, indicated here, was previously not recognised.

Notably, we observe the greatest differences between presegmented mesoderm and all somite samples, for both chromatin accessibility and differential gene expression signatures, compared to the differences seen between somites at different stages of maturation (Fig. 2d–f and Supplementary Fig. 1b and 2d–f). This emphasises the complexity of the segmentation process, when paraxial mesoderm cells transition to generate somites[3].

We also observe differential chromatin accessibility along the anterior–posterior axis in the *HOXA* cluster (Fig. 4b) as well as differential expression (Fig. 4a) and footprints (Fig. 4c–f) of *HOXA* family members. These patterns are also detected in *HOXB*, *HOXC* and *HOXD* clusters (Supplementary Fig. 4) consistent with the role of *HOX* clusters in the regionalisation of axial structures[10,11]. As mentioned above, CDX2 is highly expressed in PSM and has greater coverage of footprints in PSM compared to somite samples (Fig. 3a–d) consistent with its role in posterior axis elongation[16]. PSM samples correspond to thoracic axial levels and this region is defined by central HOX genes, which are regulated by CDX proteins[15,17]. CDX2 footprints were found in intergenic accessible chromatin peaks within the *HOXA* cluster (Fig. 4b).

Predicting enhancer gene interactions remains challenging, although new computational methods are becoming available. For example, the recent activity-by-contact model indicates that very long-range interactions are rare[59,60]. Thus, in our footprint analysis we selected accessible chromatin regions within 10 kb of the transcription start site, or within 10 kb downstream of the gene. This approach combined with experimental validation and time-lapse imaging in gastrula-stage chick embryos identified CREs for *TCF15* and *MEOX1*, both of which are in close proximity of the transcription start site (Figs. 5 and 6). For *TCF15* we identified two separate elements, *TCF15* Enh-1 and *TCF15* Enh-2, which are both active in presegmented mesoderm and somites. *TCF15* Enh-1 shows ectopic activity in the notochord. This expression was not seen when both elements were combined, suggesting that *TCF* Enh-2 may contain a repressor of notochord expression. However, both *TCF* Enh-2 and the combined CREs show ectopic activity in LPM, indicating additional repressive elements are missing. Alternatively, this CRE may interact with other gene(s) and direct their expression in the LPM. Footprint analysis identifies a RARA binding site as a highly relevant candidate TF binding site. Citrine activity is lost after mutation of the RARA site. Furthermore, dCas9-Krab epigenome modification[52] leads to loss of *TCF15* expression. Although this is restricted to the region of the embryo that is targeted by electroporation at gastrula stages[1,2], this finding suggests the endogenous CRE is essential. This element is not conserved at sequence level and might be chick specific. In contrast, the *MEOX1* enhancer identified here is conserved in avians, reptiles and mammals but not in amphibian or fish (Fig. 7a). In both chick and human CREs, TF binding sites for FOXO1 and ZIC3 are required for Citrine expression (Figs. 6e and 7c). In vivo epigenome modification of the *MEOX1* enhancer causes axial elongation phenotypes in embryos (Fig. 6g, h). RT-qPCR shows that genes involved in

chondrogenesis and sclerotome polarity are affected in somites where expression of *MEOX1* is lost after targeting the dCas9-Krab repressor to the enhancer (Fig. 6j). Furthermore, the elongation phenotypes observed are consistent with mouse mutants[21,23] and human Klippel-Feil patients who display skeletal abnormalities[25,26]. Thus, it could be of interest to determine whether the *MEOX1* enhancer is affected in patients, who do not have a coding mutation in MEOX1.

Taken together we provide a resource of paraxial mesoderm samples across a spatiotemporal series. Our analysis focussed on PPI networks and CREs important for vertebrate anterior–posterior axis formation. We assess evolutionary conservation and validate in vivo function to establish proof-of-principle, which underpins further interrogation and mining of this comprehensive data set.

## Methods

**Chicken embryos.** Fertilised chicken eggs (Henry Stewart & Co.) were incubated at 37 °C with humidity. Embryos were staged according to Hamburger and Hamilton[27]. All experiments were performed on chicken embryos younger than 14 days of development and therefore were not regulated by the Animal Scientific Procedures Act 1986.

**Embryo dissection.** HH14 embryos were dissected into Ringers solution in silicon lined petri dishes and pinned down using the extra-embryonic membranes. Ringers solution was replaced with Dispase (1.5 mg/ml) in DMEM 10 mM HEPES pH7.5 at 37 °C for 7 min prior to treatment with Trypsin (0.05%) at 37 °C for 7 min. The reaction was stopped with Ringers solution with 0.25% BSA. The PSM, ES, MS and DS were carefully dissected away from neural and lateral mesoderm tissue using sharp tungsten needles.

**RNA extraction, library preparation and sequencing.** For ES, MS and DS, consecutive four somites were dissected. Tissues were placed into RLT lysis buffer. RNA was extracted using Qiagen RNAeasy kit (Cat no. 74104) and DNase treated (Qiagen Cat no 79254) for removal of DNA. Libraries were prepared and sequenced on the Illumina HiSeq4000 platform (75 bp paired end) at the Earlham Institute. A minimum of three biological replicates for each stage were used for analysis.

**ATAC, library preparation and sequencing.** PSM, ES, MS and DS samples were dissected as stated above. Cell dissociation was performed using a protocol adapted from[50]. Briefly, tissues were dissociated with Dispase at 37 °C for 15 min with intermittent pipetting to attain a single cell suspension with 0.05% Trypsin at 37 °C for a final 5 min at 37 °C. The reaction was stopped, and cells were re-suspended in Hanks buffer (1X HBSS, 0.25% BSA, 10 mM HEPES pH8). Cells were centrifuged at 500 × g for 5 min at 4 °C, re-suspended in cold Hanks buffer, passed through 40 μm cell strainers (Fisher Cat no. 11587522), and further centrifuged at 500 × g for 5 min at 4 °C. Pelleted cells were re-suspended in 50 μl Hanks buffer, kept on ice and processed for ATAC library preparation. ATAC was performed using a protocol adapted from[31,50]. Briefly, cells were lysed in cold lysis buffer (10 mM Tris-HCl, Ph7.4, 10 mM NaCl, 3 mM MgCl2, 0.1% Igepal) and tagmentation performed using Illumina Nextera DNA kit (FC-121-1030) for 30 min at 37 °C on a shaking thermomixer. Tagmented DNA was purified using Qiagen MinElute kit (Cat no. 28004) and amplified using NEB Next High-Fidelity 2X PCR Mast Mix (Cat no. M0543S) for 11 cycles as follows: 72 °C, 5 min; 98 °C, 30 s; 98 °C, 10 s; 63 °C, 30 s; 72 °C, 1 min. Library preparation was complete after further clean up using Qiagen PCR MinElute kit (Cat no. 28004) and Beckman Coulter XP AMPpure beads (A63880). Tagmentation size was assessed using Agilent 2100 Bioanalyser. Libraries were quantified with Qubit 2.0 (Life Technologies) and sequenced using paired-end 150 bp reads on the Illumina HiSeq4000 platform at Novogene UK. Three biological replicates for each stage were used for analysis.

**Enhancer cloning.** Chick genomic DNA (gRNA) was extracted from HH14 embryos using Invitrogen Purelink gDNA extraction kit (Cat no. K1820-00). Human genomic DNA was isolated from HeLa cells. Putative enhancers were amplified using primers with specific sequence tails to enable cloning into reporter vector using a modified GoldenGate protocol[61] under the following conditions: 94 °C, 3 min; 10 cycles of 94 °C, 15 s; 55 °C, 15 s; 68 °C, 3 min, 25 cycles of 94 °C, 15 s; 63 °C, 15 s; 68 °C, 3 min; and final step of 72 °C, 4 min. Amplicons were purified using Qiagen PCR Cleanup (Cat no. 28104) and pooled with pTK nanotag reporter vector with T4 DNA ligase (Promega) and BsmBI (NEB) restriction enzyme. This reaction was prepared for T4-mediated ligation and BsmBI digestion under the following conditions: 25 cycles of 37 °C, 2 min; 16 °C, 5 min; a single step of 55 °C, 5 min; and a final step of 80 °C, 5 min. For mutagenesis of specific sites in enhancers we utilised FastCloning methodology[62].

**CRISPR-mediated enhancer repression.** sgRNAs specific for *MEOX1* Enh and *TCF15* Enh-2, or a scrambled control were cloned into a chicken pU6-3 vector using standard protocols[52]. For enhancer repression, sgRNAs and dCas9-Krab were electroporated ex ovo[52]. All primer sequences are detailed in Supplementary Table 1.

**RT-qPCR.** cDNA was synthesised from 500 ng of RNA using a Maxima First Strand cDNA synthesis kit (Thermo Fisher Scientific). qPCR was performed on a 7500 Fast Real Time PCR machine (Applied Biosystems) using SYBR Green PCR Master Mix (Thermo Fisher Scientific) according to the manufacturer's instructions. Primers (see Supplementary Table 1) were designed with Primer3Plus software (https://primer3plus.com/cgi-bin/dev/primer3plus.cgi). RT-qPCR was normalised to Gapdh mRNA. Three independent experiments each with replicate samples were performed for each RT-qPCR. The delta-delta CT[63] method was used to analyse gene expression levels. Statistical analysis was performed using GraphPad Prism (Version 6) software. Mann–Whitney non-parametric two-tail testing was applied to determine *p* values.

**Embryo preparation and ex ovo electroporation.** Hamburger and Hamilton (HH3+) embryos were captured using the filter paper based easy-culture method. Briefly, eggs were incubated for ~20 h, a window was created using forceps, the embryo and yolk were transferred into a dish and thin albumin above and around the embryo was removed using tissue paper. A circular filter paper ring was placed on top, excised transferred into a separate dish containing Ringers solution and excess yolk was removed. The embryo was then transferred into a dish containing albumin-agar and ready for electroporation with the ventral side up[49]. Plasmid DNA was injected between the membrane and embryo to cover the whole epiblast, electroporated used five pulses of 5 V, 50 ms on, 100 ms off. Thin albumin was used to seal the lids of dishes and embryos were cultured at 37 °C with humidity to the desired stage.

**Cryosectioning and immunostaining.** Embryos were fixed in 4% paraformaldehyde (PFA) for 2 h at room temperature (RT) or at 4 °C overnight, washed 3 × 10 min in PBS. Embryos were transferred into 30% sucrose/PBS overnight at 4 °C prior to 3 × 10 min washes in OCT before final embedding of OCT in dry ice. Cryosectioning was performed at 15 μm thickness. Sections were washed in 3 × 15 min PBS and 1 × 15 min in PBS/0.5% Triton X-100 prior to blocking in 5% goat serum and 5% BSA in PBS for 1 h at RT. Incubation with primary antibody for rabbit anti-GFP (1:200, Torrey Pines Biolabs Cat no. TP401) at 4 °C overnight, followed by 3 × 10 min washes in PBS and incubation with secondary antibody AlexaFluor-568-conjugated donkey anti-rabbit IgG (1:500, Thermo Fisher Cat no. A21206) for 1 h at RT. Sections were washed 3 × 10 min in PBS and 1x wash with PBS and DAPI (Sigma-Aldrich) at 0.1 mg/ml in PBS.

**Wholemount in situ hybridisation.** Wholemount in situ hybridisation using DIG-UTP labelled antisense RNA probes for *MEOX1* (a gift from Baljinder Mankoo, King's College London UK) and *TCF15* (a gift from Susanne Dietrich, University of Portsmouth UK) was carried out using standard methods. Briefly, following fixation in 4% PFA embryos were treated with Proteinase K, hybridised over night at 65 °C. After post-hybridisation washes and blocking with BMB (Roche), embryos were treated with anti-DIG antibody, coupled to alkaline phosphatase (Roche). Signal was developed using NBT/BCIP.

**Live imaging of enhancer reporter.** Embryos cultured in six-well cell culture plates (Falcon) were time-lapse imaged on an inverted wide-field microscope (Axiovert; Zeiss). Brightfield and fluorescent images were captured every 6 min for 20–24 h, using Axiovision software as described in ref. [64]. At the end of the incubation, most embryos had reached stage HH10-11.

**Image analysis.** Sections were visualised on an Axioscope with Axiovision software (Zeiss). Wholemount embryos were photographed on a Zeiss SV11 dissecting microscope with a Micropublisher 3.5 camera and acquisition software or Leica MZ16F using Leica Firecam software. Live imaging datasets were analysed in FIJI/ImageJ.

**ATAC-seq processing.** Adaptors were removed from raw paired-end sequencing reads and trimmed for quality using Trim Galore! (v.0.5.0)[65] a wrapper tool around Cutadapt[66] and FastQC[67]. Default parameters were used. Quality control (QC) was performed before and after read trimming using FastQC (v.0.11.6)[67] and no issues were highlighted from the QC process. Subsequent read alignment and post-alignment filtering was performed in concordance with the ENCODE project's "ATAC-seq Data Standards and Prototype Processing Pipeline" for replicated data (https://www.encodeproject.org/atac-seq/). In brief, reads were mapped to the chicken genome galGal5 assembly using bowtie2 (v.2.3.4.2)[68]. The resultant Sequence Alignment Map (SAM) files were compressed to the Binary Alignment Map (BAM) version on which SAMtools (v.1.9)[69] was used to filter reads that were unmapped, mate unmapped, not primary alignment or failing platform quality checks. Reads mapped as proper pairs were retained. Multi-mapping reads were

removed using the Python script assign_multimappers provided by ENCODE's processing pipeline and duplicate reads within the BAM files were tagged using Picard MarkDuplicates (v.2.18.12) [http://broadinstitute.github.io/picard/] and then filtered using SAMtools. For each step, parameters detailed in the ENCODE pipeline were used. From the processed BAM files, coverage tracks in bigWig format were generated using deepTools bamCoverage (v 3.1.2)[70] and peaks were called using MACS2 (v.2.1.1)[71] (parameters -f BAMPE -g mm -B -nomodel -shift -100 -extsize 200). Coverage tracks and peaks (narrow peak format) were uploaded to the UCSC Genome Browser[72] as custom tracks for ATAC-seq data visualisation.

**Differential accessibility and footprinting**. Analysis of ATAC-seq for differential accessibility was carried out in R (v.3.5.1)[73] using the DiffBind package (v.2.8.0)[32,33] with default parameter settings. Differential accessibility across samples was calculated using the negative binomial distribution model implemented in DEseq2 (v1.4.5)[74]. Computational footprinting analysis was conducted across samples using HINT-ATAC which is part of the Regulatory Genomic Toolbox (v.0.12.3)[42] also using default parameter settings and the galGal5 genome.

**RNA-seq differential expression analysis**. Adaptors were removed from raw paired-end sequencing reads and trimmed for quality using Trim Galore! (v.0.5.0) using default parameters. QC was performed before and after read trimming using FastQC (v.0.11.6) and no data quality issues were identified after checking the resultant QC reports. Processed reads were mapped to galGal5 cDNA using kallisto (v.0.44.0)[75]. Resultant quantification files were collated to generate an expression matrix. Differential expression, GO term and pathway analyses were then conducted using DESeq2[74] and default settings within the iDEP (v.9.0)[76] web interface. GO term analysis used PGSEA method for GO Biological Process with a minimum of 15 and maximum of 2000 geneset and <0.2 FDR. For STRING analysis, version 11.0 was used[77] to identify PPI networks with a high threshold (0.700) selected for positive interactions between pairs of genes.

**Xenopus embryo microinjection**. All experiments were carried out in accordance with relevant laws and institutional guidelines at the University of East Anglia, with full ethical review and approval, compliant to UK Home Office regulations. To obtain *Xenopus laevis* embryos, females were primed with 100 units of PMSG and induced with 500 units of human chorionic gonadotrophin. Eggs were collected manually and fertilised in vitro. Embryos were de-jellied in 2% L-cysteine, incubated at 18 °C and microinjected in 3% Ficoll into 1 cell at the 2 cell stage in the animal pole with 5 nl of enhancer reporter plasmid at 400 ng/μl or GFP capped RNA as control. Embryos were left to develop at 23 °C. Embryo stageing is according to Nieuwkoop and Faber normal table of Xenopus development. GFP capped RNA for injections was prepared using the SP6 mMESSAGE mMACHINE kit, 5 ng was injected per embryo.

**Reporting summary**. Further information on research design is available in the Nature Research Reporting Summary linked to this article.

## Data availability
The authors declare that all data supporting the findings of this study are available within the article and its supplementary information files or from the corresponding author upon reasonable request. Raw sequencing data for this study have been deposited in Sequence Read Archive (SRA) under the BioProject accession code: PRJNA602335.

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

## Acknowledgements

We thank all members of the Münsterberg and Wheeler labs for helpful discussion. Dr Timothy Grocott for discussions, Ronce Saputil and undergraduate project students for assistance with enhancer analysis. G.F.M., L.F., E.M. and V.M.H. were supported by BBSRC project grant (BB/N007034/1) to A.E.M. and G.N.W., and MRC project grant (MR/R000549/1) to A.E.M.; S.A.W. and A.M.G. were supported by studentships funded by the UKRI Biotechnology and Biological Sciences Research Council Norwich Research Park Biosciences Doctoral Training Partnership to A.E.M. and G.N.W.

## Author contributions

G.F.M. and A.E.M. conceived and designed the study. G.F.M. generated and analysed RNA-seq and ATAC-seq data, performed and analysed chick reporter expression assays, in situ hybridisation, immunohistochemistry, QPCR experiments and assisted in bioinformatic analysis. L.F. performed bioinformatic analysis on RNA-seq and ATAC-seq data. E.M. and S.A.W. assisted in live imaging, chick embryo injections and in situ hybridisation. V.M.H. assisted with cloning reporter constructs. R.M.W. and T.S.S. shared electroporation setup, plasmids and expertise in NGS. A.M.G. performed injections into *Xenopus laevis* and in situ hybridisation. G.N.W. discussed and supervised *Xenopus laevis* experiments. S.M. helped oversee the computational analysis. G.F.M. and A.E.M. discussed ideas and interpretation of data and wrote the manuscript with input from all authors. A.E.M. supervised the study.

## Competing interests

The authors declare no competing interests.
