## [Peer Review File · Nature Communications]

Reviewers' Comments:

Reviewer #1:

Remarks to the Author:

Title: "Characterising open chromatin identifies novel cis-regulatory elements important for paraxial mesoderm formation and axis extension"

Summary:

In this manuscript Mok et al. embark in a comprehensive characterization of gene expression and epigenetic regulation of gene transcription, during paraxial mesoderm differentiation along the antero-posterior axis of chicken embryos. They perform genome wide interrogation of gene expression using RNA-seq and chromatin accessibility using ATAC-seq, seeking the integration of epigenetic and transcriptomic profiling to shed light into the developmental genetic programs underlying major transitions in cell differentiation during somitogenesis and somite differentiation.

Isolation of cell populations at different axial levels in HH14 chicken embryos allowed the identification of differentially expressed genes and differential levels of chromatin accessibility between presomitic mesoderm (PSM), epithelial somites (ES), maturing somites (MS) and differentiated somites (DS). Functional analyses of differentially expressed genes and open chromatin regions was feed into overrepresentation analyses to detect the enrichment in ontology terms as well as motif representation for selected transcription factors. Despite the richness and significant volume of data generated in this broad spectrum experimental design, these analyses were used in a rather confirmatory fashion, providing moderated new insights on the gene regulatory logic controlling cell differentiation and morphogenesis along the antero-posterior axis. Consequently, the analytical workflow was guided mainly into validation of the data leaving little room for novelty, which probably left many interesting aspects of gene regulation during paraxial mesoderm differentiation uncovered. Putative non-coding regulatory elements derived from the ATAC-seq analysis near the transcription start site of *Tcf15* and *Meox1* were studied for their ability to drive the expression of fluorescent reporters in vivo. Three of these elements recapitulated in large degree the endogenous transcription pattern of the neighboring genes (*Tcf15* or *Meox1*), the ones that were likely being regulated by each of these non-coding elements. Site directed mutagenesis of predicted transcription factor binding sites (in reporter vectors) in those putative regulatory elements demonstrated these motifs to be critical for the expression of the reporter. Finally, the use of CRISPR interference to target these non-coding elements provided a valuable technique to probe the actual relevance of these elements in the control of gene transcription in vivo. These case examples not only provide evidence for these sequences to be functional (driving transcription), and the identity of the transcription factors acting on these putative binding sites provide interesting insights about how these developmental programs are regulated at the level epigenetic control of gene transcription. That part of the paper was very interesting but I think that it would have gone along the novelty vein if the authors would have explored these interactions further, specially focusing in the context of gene regulatory networks and cell signaling control of gene expression along the axis.

In general I think the paper fail in generating new insights partially because it fails in distilling out more interesting patterns from the data, in order to gain a better understanding of the developmental programs that govern paraxial mesoderm differentiation. Given the broad cell population spectra, including neural crest cell carryovers, I wish to see an analysis that goes beyond known and relatively well characterized cell differentiation phenomena (i.e. well known gene transcripts marking specific somite compartments). A more unbiased analysis would better suit this data set, but not in the spirit of just validating the approach or the fitness of the data sets to prior knowledge, but rather in spirit of discovering and exposing new patterns of developmental gene networks, thus with the expectation of using this data to illuminate not yet recognized or poorly understood aspects of paraxial mesoderm development.

Specific points:

- The authors should explain whether and how neural crest cell contamination could be a confounding factor in the analysis, and in what way this could hinder the interpretation of the data.
- Figure 1B, is not too revealing, the description in the text should suffice.
- Figure 1C, the insets to zoom in a subset of positive log2 fold change data points are confusing, remove Fig 1B and make Fig 1C larger.
- Fig 1I "z-score" scale missing?
- Fig 1J-K, there are no labels
- The section describing transcription factor footprinting reflects a major criticism of the paper, instead of using it as a validation tool, maybe it should point into exploring underappreciated regulatory networks operating along the paraxial mesoderm axis. Is this work telling us something new about how CDX2, TWIST2 or LEF1 could regulate cell fate or specific gene networks during paraxial mesoderm differentiation?
- Figure 3, TF footprinting figures have no labels on axes.
- All over the paper chromatin accessibility peaks are treated equally, besides Figure 2 G showing their distribution over introns, exons or intergenic regions. It would be informative to get a picture on what fraction of these peaks are actually transcription start sites (TSS) rather than long distance enhancer elements. Is this the situation in Figure 4 with HoxA genes? It looks like most if not all the open chromatin peaks selected overlap with TSS and no with enhancer elements. I think the authors should explore what else can we learn from the dynamics of non TSS cis-elements across the HoxA cluster? Besides patterns that seems to be synonymous with transcriptional activation at the promoter level?
- Figure 4, peaks are barely visible since the entire cluster is displayed, if possible it would be better visualized if each peak is parsed out and displayed separately.
- Figure 5, authors should explain why reporter expression is restricted only to the anterior PSM domain despite chromatin seems equally accessible in all stages.
- Authors should consider if, is it possible that TCF15 enh-1 and enh-2 have repressor sequences to inhibit expression in the lateral plate (enh-1) and the notochord (enh-2), which in synergy (Enh-1 and Enh-2) could generate the normal endogenous Tcf15 expression in vivo, excluding the ectopic expression at LPM and notochord?
- Authors should explain why the reporter is expressed in the PSM despite chromatin is not accessible at the PSM stage (Fig 6A)
- Fig 6 L, I am not entirely convinced about the expression of the reporter in PSM domain, it would be valuable to have stage match in situ hybridizations for xenopus meox1 to compare.
- The authors should consider further characterization of these candidate regulatory elements for specific DNA protein interactions using gel binding assays (EMSA) or plasmon resonance energy transfer.

Reviewer #2:

Remarks to the Author:

In this manuscript Mok and collaborators describe the transcriptomic profile as well as the chromatin accessibility profile of the paraxial mesoderm in chicken through its anteroposterior axis. Four different populations of cells have been characterized from cells placed in the presomitic mesoderm to cells from the epithelial somites, the maturing somites and differentiated somites. Both, transcriptomics and ATAC-seq approaches show clear differences among these four populations, which are related to their differentiation state as well as its anteroposterior position in the embryo. Research of differential transcription factor footprints focussing on some key TFs also show how TFs function at the four different somatic populations, also show more binding sites at the somitic population where they realize their function. Then, the authors, make a close-up view on the Hox cluster showing how both transcriptome and chromatin accessibility parallel the relative position of the somitic population within the AP axis. And finally, Mok and collaborators validated some paraxial mesoderm enhancers by focussing on two known paraxial mesoderm TFs, TCF15 and MEOX. Through a functional study in the chicken embryo, they establish two functional enhancers of TCF15 and one in MEOX. In the three cases, they demonstrate the role of different TF

binding sites present in these enhancers.

The manuscript is correctly written, it is clear and interesting for researchers interested in mesoderm development in vertebrates. For these reasons I accept this paper for publication in Nat Comm after some minor corrections that I detail below.

Minor comments/corrections:

- Concerning the co-expression clustering, the authors say that they found 4 clusters but they only present two of them. They should present all the clusters (possible as Supp material), and discuss also the two other clusters
- In page 8 there is a mistake. When talking about TF binding sites enriched in the ATACseq peaks, the authors find the binding motif for the retinoic acid receptor alpha, but they write RXRA. Since they explain that this TF can act as activator or repressor depending on the context (missing references), I imagine that it is RXRA and not RARA which is activator always in presence of RA (while it represses in the absence of RA). Please correct and explain which motif is detected, RXR or RAR.
- In the identification of differential TF footprints the authors show in Fig 3 the graphs of the different footprints for the chosen TFs. I do not know very well Hint-ATAC, but would it be possible to quantify the differential presence of a given binding site between two ATACseq samples, instead of just saying "it is more present here than there"?
- The same quantitative comment can be applied to the Hox locus section. When the authors state that accessible chromatin is "reduced", how did they calculate this reduction? It was just by "eye", or they used a quantitative statistic method? Please explain
- In the functional study of Tcf15 and MEOX section, how the authors chose the peaks they studied? For example, at the Tcf15 locus I can see several other ATAC peaks that have not been studied. Does it mean that all the peaks were tested but only the two presented showed enhancer activity? Or only two were chosen randomly and the authors just were lucky and cloned the ones that are active?
- I do not understand why the authors used a "peak approach" for Tcf15 but an evolutionary approach for MEOX. Did the authors started with the evolutionary conservation approach for both, but they only present it for MEOX because Tcf15 does not show conservation with other vertebrates at the sequence level of the two enhancers?
- Since it is well known that the sequence and position of a given enhancer are not strongly conserved, but their function usually is conserved (as shown for the MEOX enhancer in Xenopus), would it be possible to test the two Tcf15 enhancers in Xenopus also?

Reviewer #3:

Remarks to the Author:

By performing RNA-seq and ATAC-seq, the authors characterized the gene expression and accessible chromatin landscapes in paraxial mesoderm at different stages of chick embryos, presomitic mesoderm (PSM), epithelial somites (ES), maturing somites (MS), and differentiated somites (DS). They found that chromatin accessibility correlates well with gene expression in most cases and precedes the detection of transcripts in some cases. Transcription factors that regulate PSM-somite differentiation exhibit different footprints depending on the stages of paraxial mesoderm development. Furthermore, chromatin accessibility of the HoxA gene cluster changes along the anterior-posterior axis. Finally, the authors validated the function of some open chromatin regions by reporter assay using chick embryos and by introducing the sequence-specific repressors and identified novel cis-regulatory elements important for paraxial mesoderm development.

The authors successfully identified enhancer regions that regulate the expression of TCF15 and MEOX1, two factors that regulate paraxial mesoderm specification and somite patterning. This is an important study revealing the significance of the chromatin accessibility in gene regulation during paraxial mesoderm development. However, while the authors validated the enhancer sequence by using the CRISPR-mediated repression method, it does not necessarily mean that

such regions are functionally important for the endogenous gene expression. Specific comments are indicated below.

1. As stated above, the CRISPR-mediated repression targeting specific regions does not necessarily mean that such regions are functionally important for the endogenous gene expression (Figs. 5 and 6). For this experiment, the authors should test CRISPR-mediated repression targeting closed chromatin regions, rather than using scrambled gRNA, as a negative control. Alternatively, they should test CRISPR/Cas9-mediated deletion of the putative enhancer regions and analyze how the endogenous gene expression is affected in embryos.
2. In Fig. 4, the authors identified differential footprints for HoxA cluster. Some of them seem to be involved in the A-P patterning, and the activity of such regions should be tested by Citrine reporter assay using chick embryos.

We thank the reviewers for their detailed comments and are pleased that they found the work of considerable potential interest. We have considered all their comments and address all points in detail in our response.

Whilst we have not been able to do all of the experiments suggested, in part due to COVID19 lock-down and ongoing limited lab access, we feel that we have addressed the major concerns. In particular, using unbiased analysis, we have distilled out novel patterns from the data, which provides a better understanding of the developmental programs that govern paraxial mesoderm differentiation.

We are exposing new patterns of developmental gene networks. In particular, we have further explored how CDX2 and LEF1 regulate specific protein-protein interaction networks during paraxial mesoderm differentiation. This has uncovered that CDX2 and LEF1 form discrete gene networks, despite both being involved in axis elongation. The finding that the networks are largely non-overlapping suggests that parallel mechanisms drive posteriorization.

Furthermore, we have assessed the effect of loss of MEOX1 enhancer activity on endogenous genes. Overall, we believe that our work does illuminate novel aspects of paraxial mesoderm development

We highlight all changes in the manuscript text file.

Detailed response:

Reviewer #1

In this manuscript Mok et al. embark in a comprehensive characterization of gene expression and epigenetic regulation of gene transcription, during paraxial mesoderm differentiation along the antero-posterior axis of chicken embryos. They perform genome wide interrogation of gene expression using RNA-seq and chromatin accessibility using ATAC-seq, seeking the integration of epigenetic and transcriptomic profiling to shed light into the developmental genetic programs underlying major transitions in cell differentiation during somitogenesis and somite differentiation.

Isolation of cell populations at different axial levels in HH14 chicken embryos allowed the identification of differentially expressed genes and differential levels of chromatin accessibility between presomitic mesoderm (PSM), epithelial somites (ES), maturing somites (MS) and differentiated somites (DS). Functional analyses of differentially expressed genes and open chromatin regions was feed into overrepresentation analyses to detect the enrichment in ontology terms as well as motif representation for selected transcription factors.

Despite the richness and significant volume of data generated in this broad spectrum experimental design, these analyses were used in a rather confirmatory fashion, providing moderated new insights on the gene regulatory logic controlling cell differentiation and morphogenesis along the antero-posterior axis. Consequently, the analytical workflow was guided mainly into validation of the data leaving little room for novelty, which probably left many interesting aspects of gene regulation during paraxial mesoderm differentiation uncovered.

Response: We have used stringent validation in order to confirm the quality of the data set. Any novel patterns can only be relied on if expected patterns are in fact also observed. However, in addition to finding the anticipated changes consistent with axial patterning, we include new data and emphasise more the novel aspects of the study.

Putative non-coding regulatory elements derived from the ATAC-seq analysis near the transcription start site of Tcf15 and Meox1 were studied for their ability to drive the expression of fluorescent reporters in vivo. Three of these elements recapitulated in large degree the endogenous transcription pattern of the neighboring genes (Tcf15 or Meox1), the ones that were likely being regulated by each of these non-coding elements. Site directed mutagenesis of predicted transcription factor binding sites (in reporter vectors) in

those putative regulatory elements demonstrated these motifs to be critical for the expression of the reporter. Finally, the use of CRISPR interference to target these non-coding elements provided a valuable technique to probe the actual relevance of these elements in the control of gene transcription in vivo. These case examples not only provide evidence for these sequences to be functional (driving transcription), and the identity of the transcription factors acting on these putative binding sites provide interesting insights about how these developmental programs are regulated at the level epigenetic control of gene transcription. That part of the paper was very interesting but I think that it would have gone along the novelty vein if the authors would have explored these interactions further, specially focusing in the context of gene regulatory networks and cell signaling control of gene expression along the axis.

Response: Thank for these supportive comments and for appreciating the novelty of this aspect. We have now added more information and new experimental data as requested (see below).

In general I think the paper fail in generating new insights partially because it fails in distilling out more interesting patterns from the data, in order to gain a better understanding of the developmental programs that govern paraxial mesoderm differentiation. Given the broad cell population spectra, including neural crest cell carryovers, I wish to see an analysis that goes beyond known and relatively well characterized cell differentiation phenomena (i.e. well known gene transcripts marking specific somite compartments). A more unbiased analysis would better suit this data set, but not in the spirit of just validating the approach or the fitness of the data sets to prior knowledge, but rather in spirit of discovering and exposing new patterns of developmental gene networks, thus with the expectation of using this data to illuminate not yet recognized or poorly understood aspects of paraxial mesoderm development.

Specific points:

- The authors should explain whether and how neural crest cell contamination could be a confounding factor in the analysis, and in what way this could hinder the interpretation of the data.

Response: Whilst the neural crest associated genes could give some 'noise', this did not confound our analysis. We expected to see some neural crest markers in the differentiated somite (DS) sample, however, this did not mask the musculoskeletal genes or the signature of known A-P pathways.

- Figure 1B, is not too revealing, the description in the text should suffice.

Response: We agree and have moved this information to supplemental Figure S1b

- Figure 1C, the insets to zoom in a subset of positive log2 fold change data points are confusing, remove Fig 1B and make Fig 1C larger. Response: Thank you – we have done this

- Fig 1I “z-score” scale missing? We have added the scale

- Fig 1J-K, there are no labels We have added labels

- The section describing transcription factor footprinting reflects a major criticism of the paper, instead of using it as a validation tool, maybe it should point into exploring underappreciated regulatory networks operating along the paraxial mesoderm axis. Is this work telling us something new about how CDX2, TWIST2 or LEF1 could regulate cell fate or specific gene networks during paraxial mesoderm differentiation?

Response: Thank you for this valuable input. We have now discovered underappreciated regulatory networks and have focussed on CDX2 and LEF1 footprints detected in accessible chromatin within 10kb up- or downstream of expressed genes. GO-term analysis identified the associated processes and STRING analysis revealed protein-protein interaction networks. This new data is now described on pages 7/8/9 and shown in Figures 3i-m

- Figure 3, TF footprinting figures have no labels on axes.

Response: This is described in the legend of Figure 3 when footprints are first shown: “Tn5 insertion frequency across all accessible regions containing at least one CDX2 motif, at nucleotide resolution in PSM, ES, MS and DS reveals the presence of a footprint centered on the CDX2 motif.”

- All over the paper chromatin accessibility peaks are treated equally, besides Figure 2 G showing their distribution over introns, exons or intergenic regions. It would be informative to get a picture on what fraction of these peaks are actually transcription start sites (TSS) rather than long distance enhancer elements. Is this the situation in Figure 4 with HoxA genes? It looks like most if not all the open chromatin peaks selected overlap with TSS and no with enhancer elements. I think the authors should explore what

else can we learn from the dynamics of non TSS cis-elements across the HoxA cluster? Besides patterns that seems to be synonymous with transcriptional activation at the promoter level?

Response: We have reanalysed the genome-wide distribution of chromatin accessibility peaks (Figure 2g). Interestingly, nearly half of all peaks lie within 50kb of the promoter and TSS and half are in intergenic and intron regions. This is described at the bottom of page 6. We have analysed footprints in intergenic regions of the HOXA cluster. This new information is shown in Figure 4B and described on page 9 Unfortunately, the functional characterization of these candidate elements would be outside the scope of the present study. We also show the expression profiles and genome tracks for the other HOX clusters: B, C and D in supplementary Figure 4.

- Figure 4, peaks are barely visible since the entire cluster is displayed, if possible it would be better visualized if each peak is parsed out and displayed separately.

Response: The locus is complex with many accessible peaks, a detailed investigation of those peaks and their potential CRE activity will be the subject of future investigations.

- Figure 5, authors should explain why reporter expression is restricted only to the anterior PSM domain despite chromatin seems equally accessible in all stages.

Response: The reviewer is correct, the element is accessible in all samples: PSM, ES, MS and DS. TCF15 expression is differential and increases across these samples (Figure 1j). It is not uncommon for CREs to become accessible prior to high level gene expression. This is mentioned in the introductory paragraph on page 10, where we are now also referring to TCF15. The data now shown in supplementary Figure S5A-5 was removed from Figure 2j-o.

- Authors should consider if, is it possible that TCF15 enh-1 and enh-2 have repressor sequences to inhibit expression in the lateral plate (enh-1) and the notochord (enh-2), which in synergy (Enh-1 and Enh-2) could generate the normal endogenous Tcf15 expression in vivo, excluding the ectopic expression at LPM and notochord?

Response: We thank the reviewer for this insightful comment. As requested, we have generated the Enh1/Enh2 construct and indeed find that ectopic notochord expression is no longer seen, although ectopic LPM expression remains. This is now included in Figure 5d and discussed on page 11.

- Authors should explain why the reporter is expressed in the PSM despite chromatin is not accessible at the PSM stage (Fig 6A)

Response: We have added a comment on page 12, we speculate that endogenous repressor elements are missing on the reporter construct.

- Fig 6 L, I am not entirely convinced about the expression of the reporter in PSM domain, it would be valuable to have stage match in situ hybridizations for xenopus meox1 to compare.

Response: We use in situ hybridisation for MyoD, which marks paraxial mesoderm. A side-by side comparison is shown in supplementary Figure S7b.

- The authors should consider further characterization of these candidate regulatory elements for specific DNA protein interactions using gel binding assays (EMSA) or plasmon resonance energy transfer.

Response: We thank the reviewer for this suggestion and take this on board. However, unfortunately this was outside the scope of the present study.

Reviewer #2

In this manuscript Mok and collaborators describe the transcriptomic profile as well as the chromatin accessibility profile of the paraxial mesoderm in chicken through its anteroposterior axis. Four different populations of cells have been characterized from cells placed in the presomitic mesoderm to cells from the epithelial somites, the maturing somites and differentiated somites. Both, transcriptomics and ATAC-seq approaches show clear differences among these four populations, which are related to their differentiation state as well as its anteroposterior position in the embryo. Research of differential transcription factor footprints focussing on some key TFs also show how TFs function at the four different somatic populations, also show more binding sites at the somitic population where they realize their function. Then, the authors, make a close-up view on the Hox cluster showing how both transcriptome and chromatin accessibility parallel the relative position of the somitic population within the AP axis. And finally, Mok and collaborators validated some paraxial mesoderm enhancers by focussing on two known paraxial mesoderm TFs, TCF15 and MEOX. Through a functional study in the chicken embryo, they establish two functional enhancers of

TCF15 and one in MEOX. In the three cases, they demonstrate the role of different TF binding sites present in these enhancers.

The manuscript is correctly written, it is clear and interesting for researchers interested in mesoderm development in vertebrates. For these reasons I accept this paper for publication in Nat Comm after some minor corrections that I detail below.

Response: We thank the reviewer for the succinct summary of our findings and for endorsing publication.

Minor comments/corrections:

- Concerning the co-expression clustering, the authors say that they found 4 clusters but they only present two of them. They should present all the clusters (possible as Supp material), and discuss also the two other clusters

Response: We have extended this analysis and show a heatmap and t-SNE plot for 11 clusters. Here we focus on two clusters B (197 genes) and I (220 genes), all data is being made available as indicated.

- In page 8 there is a mistake. When talking about TF binding sites enriched in the ATACseq peaks, the authors find the binding motif for the retinoic acid receptor alpha, but they write RXRA. Since they explain that this TF can act as activator or repressor depending on the context (missing references), I imagine that it is RXRA and not RARA which is activator always in presence of RA (while it represses in the absence of RA). Please correct and explain which motif is detected, RXR or RAR.

Response: Thank you we corrected the statement in the text, moved to page 8 and added a relevant review. The motif detected is annotated as rxra (Figure 2i), the footprints detected were annotated as RARA (supplementary Figure S3j-l, removed from Figure 3p)

- In the identification of differential TF footprints the authors show in Fig 3 the graphs of the different footprints for the chosen TFs. I do not know very well Hint-ATAC, but would it be possible to quantify the differential presence of a given binding site between two ATACseq samples, instead of just saying "it is more present here than there"?

Response: This is quantified, the total n of footprints is shown in each panel.

- The same quantitative comment can be applied to the Hox locus section. When the authors state that accessible chromatin is "reduced", how did they calculate this reduction? It was just by "eye", or they used a quantitative statistic method? Please explain

Response: We used DiffBind, a statistical package, for differential peak calling. All peaks are now highlighted in grey boxes for the intergenic regions.

- In the functional study of Tcf15 and MEOX section, how the authors chose the peaks they studied? For example, at the Tcf15 locus I can see several other ATAC peaks that have not been studied. Does it mean that all the peaks were tested but only the two presented showed enhancer activity? Or only two were chosen randomly and the authors just were lucky and cloned the ones that are active?

Response: Additional peaks were assessed but did not show activity usign the Citrine-reporter.

- I do not understand why the authors used a "peak approach" for Tcf15 but an evolutionary approach for MEOX. Did the authors started with the evolutionary conservation approach for both, but they only present it for MEOX because Tcf15 does not show conservation with other vertebrates at the sequence level of the two enhancers?

Response: Evolutionary conservation, if present, provides an additional selection criterium. However, this is not always the case and we now explain that the TCF15 is not conserved and could be chick specific (bottom of page 10).

- Since it is well known that the sequence and position of a given enhancer are not strongly conserved, but their function usually is conserved (as shown for the MEOX enhancer in Xenopus), would it be possible to test the two Tcf15 enhancers in Xenopus also?

Response: We examined the TCF15 enhancers in Xenopus. They did not show any activity.

Reviewer #3

By performing RNA-seq and ATAC-seq, the authors characterized the gene expression and accessible chromatin landscapes in paraxial mesoderm at different stages of chick embryos, presomitic mesoderm (PSM), epithelial somites (ES), maturing somites (MS), and differentiated somites (DS). They found that chromatin accessibility correlates well with gene expression in most cases and precedes the detection of

transcripts in some cases. Transcription factors that regulate PSM-somite differentiation exhibit different footprints depending on the stages of paraxial mesoderm development. Furthermore, chromatin accessibility of the HoxA gene cluster changes along the anterior-posterior axis. Finally, the authors validated the function of some open chromatin regions by reporter assay using chick embryos and by introducing the sequence-specific repressors and identified novel cis-regulatory elements important for paraxial mesoderm development.

The authors successfully identified enhancer regions that regulate the expression of TCF15 and MEOX1, two factors that regulate paraxial mesoderm specification and somite patterning. This is an important study revealing the significance of the chromatin accessibility in gene regulation during paraxial mesoderm development.

Response: We thank the reviewer for this concise summary of our work and for recognising its importance.

However, while the authors validated the enhancer sequence by using the CRISPR-mediated repression method, it does not necessarily mean that such regions are functionally important for the endogenous gene expression. Specific comments are indicated below.

1. As stated above, the CRISPR-mediated repression targeting specific regions does not necessarily mean that such regions are functionally important for the endogenous gene expression (Figs. 5 and 6). For this experiment, the authors should test CRISPR-mediated repression targeting closed chromatin regions, rather than using scrambled gRNA, as a negative control. Alternatively, they should test CRISPR/Cas9-mediated deletion of the putative enhancer regions and analyze how the endogenous gene expression is affected in embryos.

Response: Respectfully, we disagree with this statement. The dCas9-Krab repressor when targeted to the endogenous enhancer reports the functionality of the enhancer. We show that the endogenous expression of the linked genes, TCF15 or MEOX1 respectively, is reduced. Reduced expression is observed in the area that is targeted in the embryo by electroporation at gastrula stages.

Furthermore, we identified genes associated with a MEOX1 footprint. We focus on those involved in chondrogenesis and sclerotome polarity and show that their expression is also reduced after dCas9-Krab mediated epigenome modification of the MEOX1 enhancer (Figure 6i, j).

2. In Fig. 4, the authors identified differential footprints for HoxA cluster. Some of them seem to be involved in the A-P patterning, and the activity of such regions should be tested by Citrine reporter assay using chick embryos.

Response: We thank the reviewer for their interest in the HOX cluster and its role in A-P patterning. However, in the first instance we chose to focus on two novel CREs associated with genes important for somite differentiation. The HOX clusters show complex patterns of differential accessibility and we plan to investigate this in more detail in future. We think that the functional characterization of these candidate elements would be outside the scope of the present study.

Reviewers' Comments:

Reviewer #2:

Remarks to the Author:

I have read the new manuscript as well as the author's answer to the comments and criticisms of all reviewers, and I find that the new version has been improved.

In particular all my questions have been answered and I consider that the manuscript can be accepted for publication

Reviewer #3:

Remarks to the Author:

I have no further comments.